# SNAP25 disease mutations change the energy landscape for synaptic exocytosis due to aberrant SNARE interactions

Anna Kádková[1†], Jacqueline Murach[2†], Maiken Østergaard[1†], Andrea Malsam[2], Jörg Malsam[2], Fabio Lolicato[2,3], Walter Nickel[2], Thomas H Söllner[2], Jakob Balslev Sørensen[1]*

[1]Department of Neuroscience, University of Copenhagen, Copenhagen, Denmark; [2]Heidelberg University Biochemistry Center, Heidelberg, Germany; [3]Department of Physics, University of Helsinki, Helsinki, Finland

**Abstract** SNAP25 is one of three neuronal SNAREs driving synaptic vesicle exocytosis. We studied three mutations in SNAP25 that cause epileptic encephalopathy: V48F, and D166Y in the synaptotagmin-1 (Syt1)-binding interface, and I67N, which destabilizes the SNARE complex. All three mutations reduced Syt1-dependent vesicle docking to SNARE-carrying liposomes and $Ca^{2+}$-stimulated membrane fusion in vitro and when expressed in mouse hippocampal neurons. The V48F and D166Y mutants (with potency D166Y > V48F) led to reduced readily releasable pool (RRP) size, due to increased spontaneous (miniature Excitatory Postsynaptic Current, mEPSC) release and decreased priming rates. These mutations lowered the energy barrier for fusion and increased the release probability, which are gain-of-function features not found in *Syt1* knockout (KO) neurons; normalized mEPSC release rates were higher (potency D166Y > V48F) than in the *Syt1* KO. These mutations (potency D166Y > V48F) increased spontaneous association to partner SNAREs, resulting in unregulated membrane fusion. In contrast, the I67N mutant decreased mEPSC frequency and evoked EPSC amplitudes due to an increase in the height of the energy barrier for fusion, whereas the RRP size was unaffected. This could be partly compensated by positive charges lowering the energy barrier. Overall, pathogenic mutations in SNAP25 cause complex changes in the energy landscape for priming and fusion.

*For correspondence: jakobbs@sund.ku.dk

†These authors contributed equally to this work

Competing interest: The authors declare that no competing interests exist.

## eLife assessment

This study documents **important** findings on three variants in SNAP25 that are associated with developmental and epileptic encephalopathy. The thorough characterization of synaptic release and in vitro vesicle fusion phenotypes provides interesting information about the nature of the SNAP25 variants. The evidence supporting the claims is **compelling**, and this work will be of interest to neuroscientists working on SNAP25, SNAP25-associated encephalopathy, and synaptic vesicle exocytosis.

## Introduction

The fusion machinery responsible for chemical synaptic transmission is well known: it consists of the SNARE complex, a ternary complex formed by the proteins VAMP2, syntaxin-1, and SNAP25 (*Sutton et al., 1998*). This complex is under tight control by upstream partner protein Munc18-1, which acts as a template for SNARE complex formation, and Munc13s, which assist in the transitions required along the pathway of assembly (*Rizo, 2022*). SNARE complexes assemble in a zipper-like manner; partially

**eLife digest** Neurons in the brain communicate with one another by passing molecules called neurotransmitters across the synapse connecting them together. Mutations in the machinery that controls neurotransmitter release can lead to epilepsy or developmental delays in early childhood, but how exactly is poorly understood.

Neurotransmitter release is primarily controlled by three proteins that join together to form the SNARE complex, and another protein called synaptotagmin-1. This assembly of proteins primes vesicles containing neurotransmitter molecules to be released from the neuron. When calcium ions bind to synaptotagmin-1, this triggers vesicles in this readily releasable pool to then fuse with the cell membrane and secrete their contents into the small gap between the communicating neurons.

Mutations associated with epilepsy and developmental delays have been found in all components of this release machinery. Here, Kádková, Murach, Østergaard et al. set out to find how three of these mutations, which are found in a protein in the SNARE complex called SNAP25, lead to aberrant neurotransmitter release.

Two of these mutations are located in the interface between the SNARE complex and synaptotagmin-1, while the other is found within the bundle of proteins that make up the SNARE complex. In vitro and ex vivo experiments in mice revealed that the two interface mutations led to defects in vesicle priming, while at the same time bypassing the control by synaptotagmin-1, resulting in vesicles spontaneously fusing with the cell membrane in an unregulated manner. These mutations therefore combine loss-of-function and gain-of-function features.

In contrast, the bundle mutation did not impact the number of vesicles in the releasable pool but reduced spontaneous and calcium ion evoked vesicle fusion. This was due to the mutation destabilizing the SNARE complex, which reduced the amount of energy available for merging vesicles to the membrane.

These findings reveal how SNAP25 mutations can have different effects on synapse activity, and how these defects disrupt the release of neurotransmitters. This experimental framework could be used to study how other synaptic mutations lead to diseases such as epilepsy. Applying this approach to human neurons and live model organisms may lead to the discovery of new therapeutic targets for epilepsy and delayed development.

assembled SNAREpins bind synaptotagmin-1 (Syt1) and complexin within two separate interaction sites: the primary interface formed between a synaptotagmin-1 molecule and syntaxin-1 and SNAP25 (*Zhou et al., 2015*), and the tripartite interface formed by syntaxin-1 and VAMP2 with another Syt1 molecule and complexin (*Zhou et al., 2017*). Upon arrival of an action potential, $Ca^{2+}$ binds to the two C2 domains of Syt1, which results in rapid vesicle–plasma membrane fusion and release of neurotransmitter within a fraction of a millisecond (*Südhof, 2013*).

The strong functional integration and specialization of the neuronal SNARE for speed have rendered the release machinery exquisitely susceptible to insults. De novo mutations in SNAREs and associated proteins lead to complex neurological disease, characterized by drug-resistant epilepsy, intellectual disability, movement disorders, and often autism; a syndrome which has been denoted 'SNAREopathy' (*Verhage and Sørensen, 2020*). Although rare, these are devastating conditions for the patients and their families. Consequently, there is considerable interest in revealing the molecular/cellular mechanisms for these conditions, which is seen as key to the development of treatment. Disease-causing mutations in the SNARE-machinery fall into distinct categories according to the nature of the defect (*Verhage and Sørensen, 2020*). These include different forms of haploinsufficiency, where the mutated protein is either lost altogether or has lost its functionality, and dominant-negative or recessive variants, as well as variants with new or changed protein interactions, referred to as neomorphs.

Most SNAREopathy mutations have been described in *STXBP1* (*Abramov et al., 2021a*; *Verhage and Sørensen, 2020*; *Xian et al., 2022*), the gene encoding Munc18-1. These mutations are generally found to cause Munc18-1 hypo-expression, probably due to protein instability (*Guiberson et al., 2018*; *Kovacevic et al., 2018*; *Martin et al., 2014*; *Saitsu et al., 2008*), and thus the mutations belong in the haploinsufficiency category. Synaptic phenotypes of Munc18-1 hypo-expression have been identified in the *Drosophila* neuromuscular junction (*Wu et al., 1998*), in cultured mouse neurons

(*Toonen et al., 2006*) and in brain slices of *STXBP1* heterozygous mice (*Chen et al., 2020*; *dos Santos et al., 2023*), as well as in human neurons (*Patzke et al., 2015*), whereas expression of human missense mutations in *STXBP1* null mouse neurons led to synaptic phenotypes for some mutations, but not for others (*Kovacevic et al., 2018*). Attempts to rescue the disease phenotypes have focused on mechanisms for increasing expression levels, for instance chemical chaperones that might prevent Munc18-1 misfolding and degradation (*Abramov et al., 2021b*; *Guiberson et al., 2018*).

The situation is different for disease mutations in Syt1 and SNAP25. In these cases, mutation generally does not cause protein instability, but instead changes function in different ways. For Syt1, disease mutations were found to cluster in the C2B domain around the top loops that coordinate $Ca^{2+}$ (*Baker et al., 2018*). Expression of these mutants in neurons demonstrated reduced evoked release, in some cases an increase in spontaneous release, and a dominant-negative phenotype when co-expressed with wildtype (WT) protein (*Bradberry et al., 2020*). The molecular mechanism was identified as a decrease in $Ca^{2+}$-dependent lipid binding (*Bradberry et al., 2020*). In SNAP25, disease-causing mutations are found within the SNARE domains (*Hamdan et al., 2017*; *Klöckner et al., 2021*; *Rohena et al., 2013*; *Shen et al., 2014*). *Alten et al., 2021* studied a selection of SNAP25 mutants, and found no changes in expression levels, but changes in both spontaneous and evoked release. Specifically, mutations in the primary Syt1:SNARE interface (*Zhou et al., 2015*) caused an increase in spontaneous release rates, and a decrease in evoked release amplitudes. Conversely, C-terminal mutations in the so-called 'layer residues', whose side chains point to the center of the SNARE complex and are involved in SNAREpin zippering (*Sutton et al., 1998*), led to a decrease in both evoked and spontaneous release (*Alten et al., 2021*). This is expected because the C-terminal end of the SNARE complex is required for both types of release (*Weber et al., 2010*).

*Alten et al., 2021* described striking phenotypes for most SNAP25 mutations tested, but the molecular reason for these phenotypes remains incompletely understood, and a few findings were surprising. For instance, Alten et al. reported that the V48F and D166Y mutants supported an unchanged readily releasable pool (RRP) of vesicles. The RRP is the pool of vesicles available for immediate release upon invasion of the presynapse by the action potential, and it is often assessed by applying a hyperosmotic solution (often 0.5 M sucrose), which causes primed vesicles to fuse (*Rosenmund and Stevens, 1996*). Previously, we concluded that the primary Syt1:SNARE interface is involved in vesicle priming (*Schupp et al., 2016*), which agrees with the suggestion that Syt1 binds to the SNAREs before $Ca^{2+}$ arrival (*Zhou et al., 2015*). Conversely, Alten et al. reported a smaller RRP for the I67N mutant, but the I67 residue is present in the internal of layer +4, which we expected to affect final SNARE complex zippering causing membrane fusion rather than priming (*Sørensen et al., 2006*; *Weber et al., 2010*). These discrepancies might be explained by the fact that Alten et al. used mixed cultures and mainly studied GABAergic transmission. Glutamatergic transmission is hard to investigate in mixed cultures due to reverberating activity, but it can be studied using autaptic neurons, which was our approach (*Schupp et al., 2016*; *Weber et al., 2010*). Thus, there is a need to further study these disease mutations in glutamatergic neurons. Another open question is whether the phenotype of the mutants in the primary interface (V48F and D166Y) is explainable solely by the loss of Syt1 binding, or whether other features of these mutations add to, or detract from, the phenotype.

Here, we reexamined three SNAP25 disease mutations using glutamatergic autaptic neurons: I67N, V48F, and D166Y. We confirmed the increase in spontaneous release rate and decrease in evoked release by V48F and D166Y reported previously (*Alten et al., 2021*). Through in vitro analysis we find that both mutations lower Syt1 association to SNARE-protein liposomes. Additional experiments both in vitro and in cells demonstrate that V48F and especially D166Y represent partial gain-of-function mutations that increase association to partner SNAREs and lower the energy barrier for fusion, bypassing Syt1-dependent control. Thus, these mutants do not phenocopy the loss of Syt1, but combine loss of Syt1 binding with a gain-of-function phenotype. At the same time, the mutants act as loss-of-function in upstream reactions, through effects on priming. The I67N is a classical dominant-negative mutation, which increases the energy barrier for fusion, but does not change the size of the RRP if probed by a sufficiently high concentration of sucrose, nor does the I67N change the electrostatics of triggering itself. Thus, for V48F and D166Y, loss-of-function and gain-of-function features combine to change the energy landscape for vesicle priming and fusion in a complex way. These findings have consequences for our understanding of the simultaneous role of the primary SNARE:Syt1 interface in vesicle priming and release clamping. It further demonstrates the challenge

**Figure 1.** Localization of three pathogenic mutations in SNAP25. (**A**) Schematic of the neuronal SNARE complex interacting with C2B domain of synaptotagmin-1 (Syt1; not to scale) via the primary interface. Position of the I67N mutation in the first SNARE domain of SNAP25 is depicted by an asterisk. (**B**) Interaction site of the C2B domain of Syt1 and SNAP25. Syt1 interacts with SNAP25 both electrostatically (regions I and II) and within the hydrophobic patch (HP patch) (*Zhou et al., 2015*). (**C**) Position of the disease-linked mutations V48F (hydrophobic patch) and D166Y (region I) in the SNARE complex.

faced by finding mechanism-based treatments of these disorders in the presence of multiple effects caused by single-point mutations.

## Results

We first investigated two SNAP25 disease-causing mutations within the primary Syt1:SNARE interface (V48F and D166Y; *Figure 1A–C*). The mutations occurred de novo and were identified in single heterozygous subjects. The V48F mutant was identified in a 15-year-old female with encephalopathy, intellectual disability and generalized epilepsy with seizures started at 5 months of age; Magnetic Resonance Imaging (MRI) was normal except for delayed myelination (*Rohena et al., 2013*). The D166Y mutants were found in a 23-year-old male with global developmental delay, nocturnal tonic–clonic seizures, and moderate intellectual disability; the MRI showed mild diffuse cortical atrophy (*Hamdan et al., 2017*). We aimed to achieve a detailed understanding of the reason for synaptic dysfunction.

### The V48F and D166Y mutants disinhibit spontaneous release and desynchronize evoked release

We constructed lentiviral vectors, which expressed WT or mutant SNAP25b fused N-terminally to EGFP as an expression marker (*Delgado-Martínez et al., 2007*). In the absence of SNAP25, neuronal viability is compromised (*Delgado-Martínez et al., 2007*; *Peng et al., 2013*; *Santos et al., 2017*; *Weber et al., 2010*) with low-density autaptic glutamatergic neurons dying within a few days, whereas *Snap25* KO neurons growing at higher densities, or in the intact brain, can survive longer, but also eventually degenerate (*Bronk et al., 2007*; *Hoerder-Suabedissen et al., 2019*). We therefore first examined the morphology and survival of autaptic hippocampal neurons from *Snap25* KO after expressing mutated or WT EGFP-SNAP25b (henceforth denoted 'WT' in rescue experiments; not to be confused with '*Syt1* WT', which refers to wildtype littermates of *Syt1* KO mice).

As expected, *Snap25* KO autaptic neurons did not survive in the absence of expression of exogeneous SNAP25 (*Figure 2G*), whereas expression of WT EGFP-SNAP25b restored survival (*Delgado-Martínez et al., 2007*; *Ruiter et al., 2019*; *Weber et al., 2010*). Both mutations (V48F and D166Y) caused rescue of survival; however, the number of neurons per islet was lower than in WT expressing neurons prepared in parallel. This difference was statistically significant for D166Y, but not for V48F (*Figure 2G*). Since patients harbor one mutated and one WT allele, we co-expressed WT EGFP-SNAP25b with mutated EGFP-SNAP25b, by combining infection with separate viruses encoding WT and mutant protein in a 1:1 ratio, as done previously by others for Syt1 (*Bradberry et al., 2020*). As a prerequisite for this, we demonstrated by Western blot analysis that both mutant viruses expressed similar amounts of protein as WT viruses (*Figure 2A–C*). In co-expressing neurons, we added half the volume of WT and mutant virus as compared to the WT condition, thus keeping the total amount of virus constant. We preferred to combine two single-cistronic viruses, rather than to construct bicistronic viruses, since the larger insert would be expected to lower the titer of viruses, which

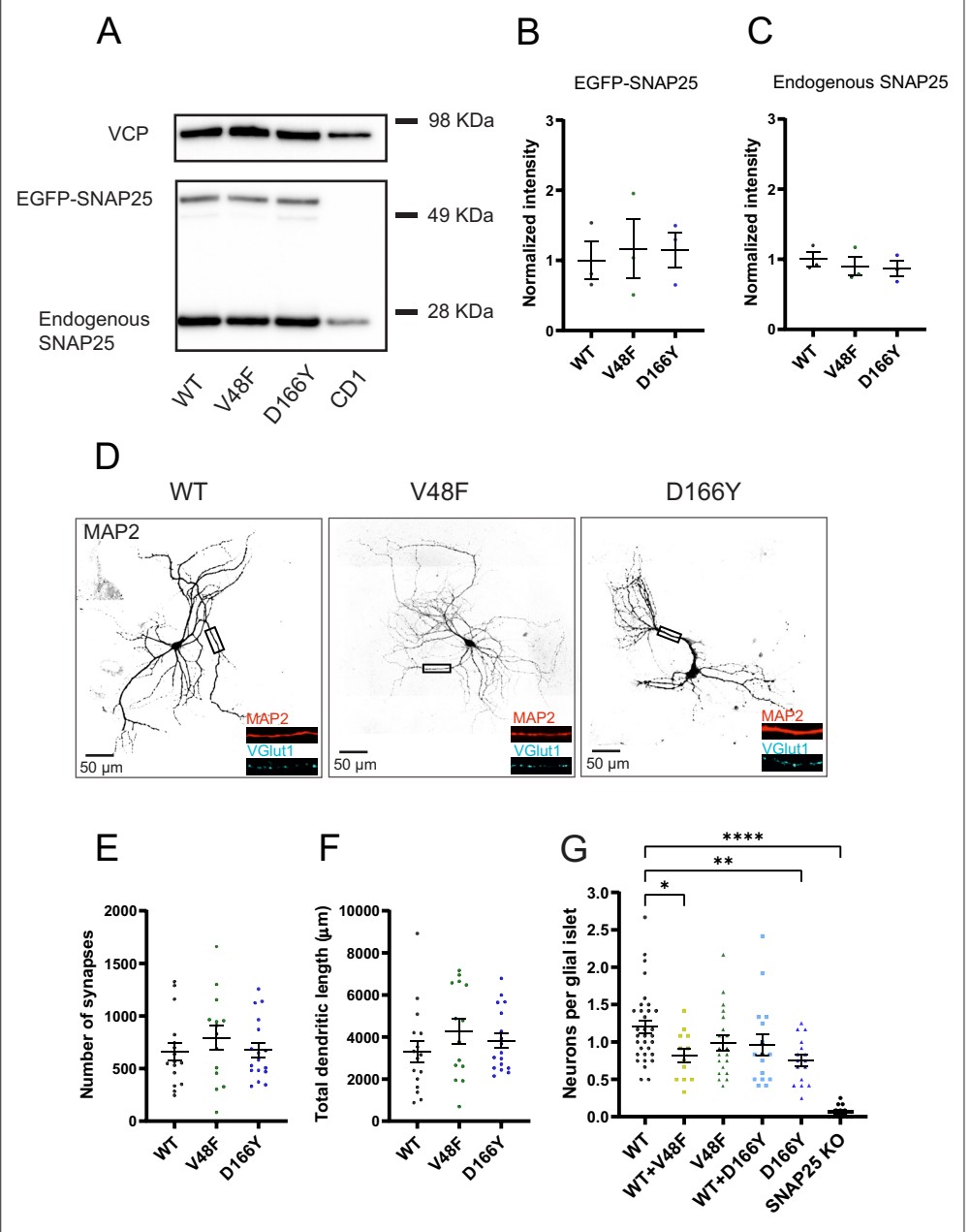

**Figure 2.** Pathogenic SNAP25 mutations compromise neuronal viability, but not synaptogenesis. (**A**) SNAP25b V48F and D166Y mutations are similarly expressed as the wildtype (WT) SNAP25b protein. EGFP-SNAP25b was overexpressed in neurons from CD1 (WT) mice; both endogenous and overexpressed SNAP25 are shown. Valosin-containing protein (VCP) was used as loading control. Quantification of EGFP-SNAP25b (**B**) and endogenous SNAP25 (**C**) from Western blots (as in A). Displayed are the intensity of EGFP-SNAP25b or endogenous SNAP25 bands, divided by the intensity of VCP bands, normalized to the WT situation (*n* = 3 independent experiments). The expression level of mutants was indistinguishable from expressed WT protein (analysis of variance, ANOVA). (**D**) Representative images of control (WT) and mutant (V48F, D166Y) hippocampal neurons stained by dendritic (MAP2) and synaptic (VGlut1) markers. Displayed is MAP2 staining, representing the cell morphology, in inserts MAP2 staining is depicted in red and VGlut staining in cyan. The scale bar represents 50 µm. (**E**) Number of synapses per neuron in WT and mutant cells. (**F**) Total dendritic length of WT and mutant neurons. (**G**) Cell viability represented as the number of neurons per glia island. ****p < 0.0001, **p < 0.01, *p < 0.05, Brown–Forsythe ANOVA test with Dunnett's multiple comparisons test.

The online version of this article includes the following source data for figure 2:

**Source data 1.** Excel file containing quantitative data.

*Figure 2 continued on next page*

*Figure 2 continued*

**Source data 2.** Original files for the Western blot analysis in **Figures 2A and 10A** (anti-SNAP25 and anti-VCP).

**Source data 3.** PDF containing **Figure 2A** and **10A**, and original scans of the Western blots with highlighted bands and sample labels.

might compromise survival rescue of *Snap25* KO neurons. Using co-expression of WT and mutant SNAP25b, neuronal survival was mildly and significantly reduced in the V48F + WT condition from WT (**Figure 2G**). Overall, mutations seemed to mildly reduce survivability of neurons. Staining of neurons against VGlut1 (a marker for glutamatergic synapses) and MAP2 (a dendritic marker) revealed no significant difference in the dendritic length or the number of synapses in neurons expressing V48F, and D166Y alone (**Figure 2D–F**).

Patch-clamp was performed on days 10–14 on autaptic neurons expressing either the WT EGFP-SNAP25b, the mutants, or both the mutant and WT in a 1:1 ratio. Spontaneous miniature events were recorded at a holding voltage of −70 mV. Both primary interface mutations (V48F and D166Y) strongly increased the frequency of mEPSCs (**Figure 3A, B, D, E**); the mEPSC amplitude was significantly increased in the V48F, and insignificantly (p = 0.14) increased for the D166Y (**Figure 3C, F**). The mEPSC frequency was at least as high as in *Syt1* KO neurons measured in separate experiments (**Figure 3G–I**). V48F and D166Y co-expressed with WT resulted in mEPSC frequencies close to the arithmetic mean between frequencies in WT and the mutant alone (**Figure 3B, E**), indicating that the mutations are incompletely dominant, or co-dominant.

Brief depolarization elicits an unclamped action potential in the axon, which makes it possible to study evoked release, which is essentially absent in *Snap25* KO neurons (**Bronk et al., 2007**; **Delgado-Martínez et al., 2007**). The V48F and D166Y mutants both supported evoked release, but the evoked EPSC (eEPSC) amplitude was significantly reduced in the mutant condition (**Figure 4A, B, E, F**), whereas when co-expressed with the WT, only the D166Y mutant had a significantly reduced eEPSC (**Figure 4B, F**). Integration of individual eEPSCs allowed determination of the total charge and the assessment of synchronous and asynchronous release components (**Figure 4—figure supplement 1**). The eEPSC charge was significantly reduced in the D166Y, but not in the V48F (**Figure 4C, G**), whereas in both cases the kinetics was significantly shifted in the direction of more asynchronous release, and the fast time constant was prolonged (**Figure 4D, H**; **Figure 4—figure supplement 2**). We compared the data obtained from V48F and D166Y SNAP25 to *Syt1* KO neurons recorded in separate experiments. *Syt1* WT neurons in this set of experiments had larger eEPSC amplitudes, and a larger charge than WT-rescued *Snap25* KO neurons (**Figure 4I–K**), which might be caused by differences between cell cultures, or animal lines (*SNAP25* vs. *Syt1* lines). Note that our experiments using WT, mutant, and WT + mutant SNAP25 were always carried out in neurons prepared and recorded in parallel from the same *Snap25* KO embryos (Materials and methods). In the *Syt1* KO, the kinetic change was similar to the V48F and D166Y (**Figure 4L**), but the total charge was also strongly reduced (**Figure 4K**), unlike the two mutations that displayed at most a mild reduction (**Figure 4C, G**).

Overall, D166Y and V48F caused a strong disinhibition of spontaneous release and a desynchronization of evoked release, as demonstrated before (**Alten et al., 2021**), and consistent with previous mutational studies of the primary interface, both in SNAP25 (**Schupp et al., 2016**) and Syt1 (**Zhou et al., 2015**). Accordingly, these phenotypes are similar to *Syt1* KO, but the effect of V48F and D166Y on total evoked charge was milder than in the *Syt1* KO. The preserved overall charge of evoked release in V48F and mild reduction in D166Y might point to a compensatory gain-of-function aspect to these mutations, in addition to the impaired Syt1 interaction.

## V48F and D166Y mutations lower the energy barrier for release and reduce the RRP size

The role of Syt1 and Syt1:SNARE interactions in vesicle priming has been controversially discussed. The RRP is often assessed by applying a pulse of hypertonic solution to the neurons, usually 0.5 M sucrose (**Rosenmund and Stevens, 1996**; **Schotten et al., 2015**). However, in some experiments this did not lead to a change in RRP size in the absence of Syt1 (**Bacaj et al., 2015**; **Xu et al., 2009**), whereas experiments both from our laboratories and others showed a decrease in RRP in the absence of Syt1 (**Bouazza-Arostegui et al., 2022**; **Chang et al., 2018**; **Courtney et al., 2019**; **Huson et al.,**

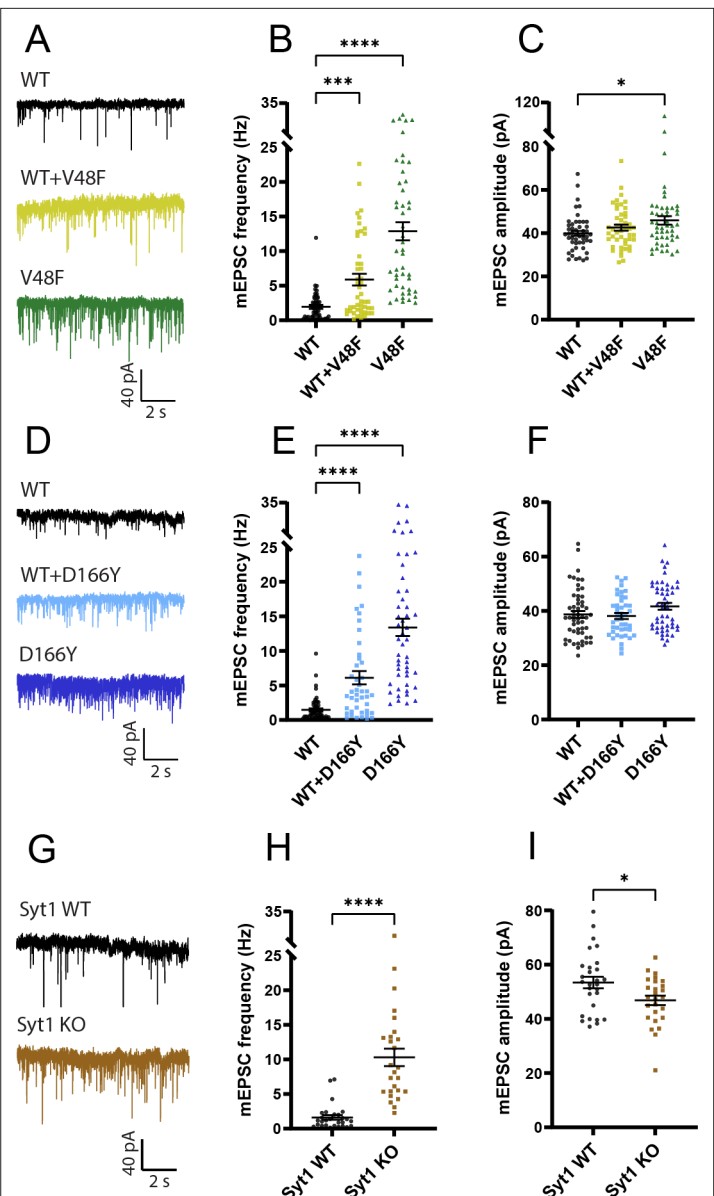

**Figure 3.** V48F and D166Y mutations increase mEPSC frequency. (**A, D, G**) Example traces of mEPSC release for wildtype (WT), mutant, and 1:1 co-expression of WT and mutant SNAP25b, or (**G**) Syt1 WT and knockout (KO). (**B, E**) The mEPSC frequencies were increased in both V48F and D166Y mutants and co-expressed conditions (V48F: *n* = 49, 47, 48 for WT, co-expressed, and mutant conditions, respectively; D166Y: n = 54, 43, 50). ****p < 0.0001, ***p < 0.001, Brown–Forsythe analysis of variance (ANOVA) test with Dunnett's multiple comparisons test. (**C, F**) mEPSC amplitudes were on average increased by the V48F and D166Y mutations; this was significant for the V48F. *p <0 .05, ANOVA with Dunnett's multiple comparison test. (**H, I**) Syt1 WT and KO data (Syt1: *n* = 28, 26 for the WT and KO condition). The mEPSC frequencies and amplitudes were increased and decreased in the KO, respectively. ****p < 0.0001, Welch's *t*-test, *p < 0.05, unpaired *t*-test.

The online version of this article includes the following source data for figure 3:

**Source data 1.** Excel file containing quantitative data.

2020; *Liu et al., 2009*; *Ruiter et al., 2019*). The reason for this discrepancy is likely partly technical and has to do with how fast sucrose can be applied to the dendritic tree (see also Discussion). Using neurons growing on small autaptic islands makes it possible to apply sucrose to the entire dendritic tree (which is confined to the 30–50 µm island) within tens of milliseconds, which is fast enough to

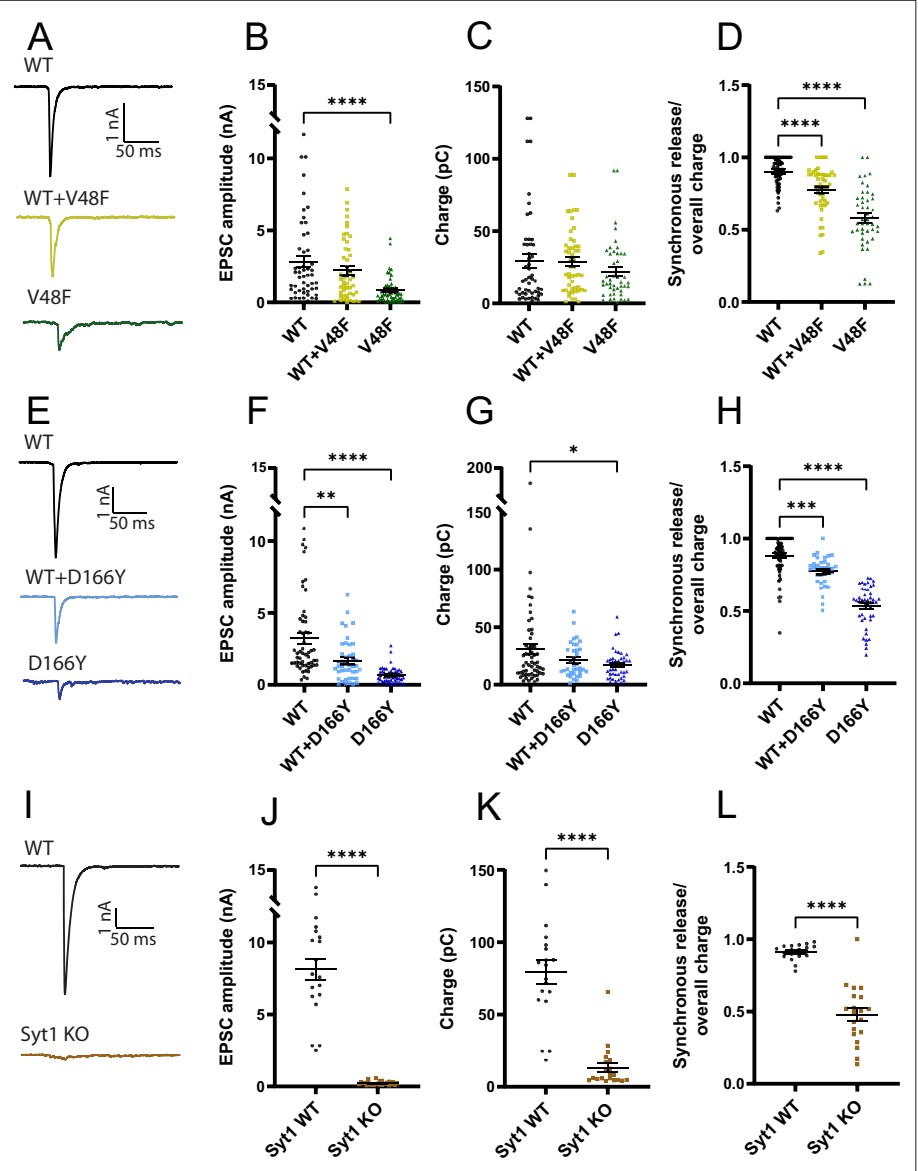

**Figure 4.** V48F and D166Y mutations reduce the amplitude of the eEPSC. (**A, E, I**) Example evoked excitatory postsynaptic currents (eEPSC) for wildtype (WT), SNAP25b mutants, and co-expressed WT/mutants, or (**I**) Syt1 WT and knockout (KO). (**B, F, J**) eEPSC amplitude was decreased by both SNAP25b mutations (V48F: $n = 50, 50, 45$ for WT, co-expressed, and mutant conditions, respectively; D166Y: $n = 56, 35, 44$) and by Syt1 KO (Syt1: $n = 19, 26$ for the WT and KO condition). SNAP25b mutations: ****$p < 0.0001$, **$p < 0.01$, Brown–Forsythe analysis of variance (ANOVA) test with Dunnett's multiple comparisons test; Syt1: ****$p < 0.0001$, Welch's $t$-test. (**C, G, K**) Overall evoked charge after a single depolarization (V48F: $n = 50, 45, 50$ for WT, mutant, and co-expressed conditions, respectively; D166Y: 56, 44, 35; Syt1: 19, 20 for WT and KO). SNAP25b: *$p < 0.05$, Brown–Forsythe ANOVA with Dunnett's multiple comparison test; Syt1: ****$p < 0.0001$, Welch's $t$-test. (**D, H, L**) Fractional contribution of the synchronous release component to the overall charge (V48F: $n = 50, 50, 45$ for WT, co-expressed, and mutant conditions, respectively; D166Y: 56, 35, 44; Syt1: 19, 20 for WT and KO). SNAP25b: ****$p < 0.0001$, ***$p < 0.001$, Brown–Forsythe ANOVA (V48F) or standard ANOVA (D166Y) with Dunnett's multiple comparisons test; Syt1: ****$p < 0.0001$, Welch's $t$-test.

The online version of this article includes the following source data and figure supplement(s) for figure 4:

**Source data 1.** Excel file containing quantitative data.

**Figure supplement 1.** Kinetic parameters of evoked EPSCs.

**Figure supplement 2.** Kinetic parameters of eEPSCs.

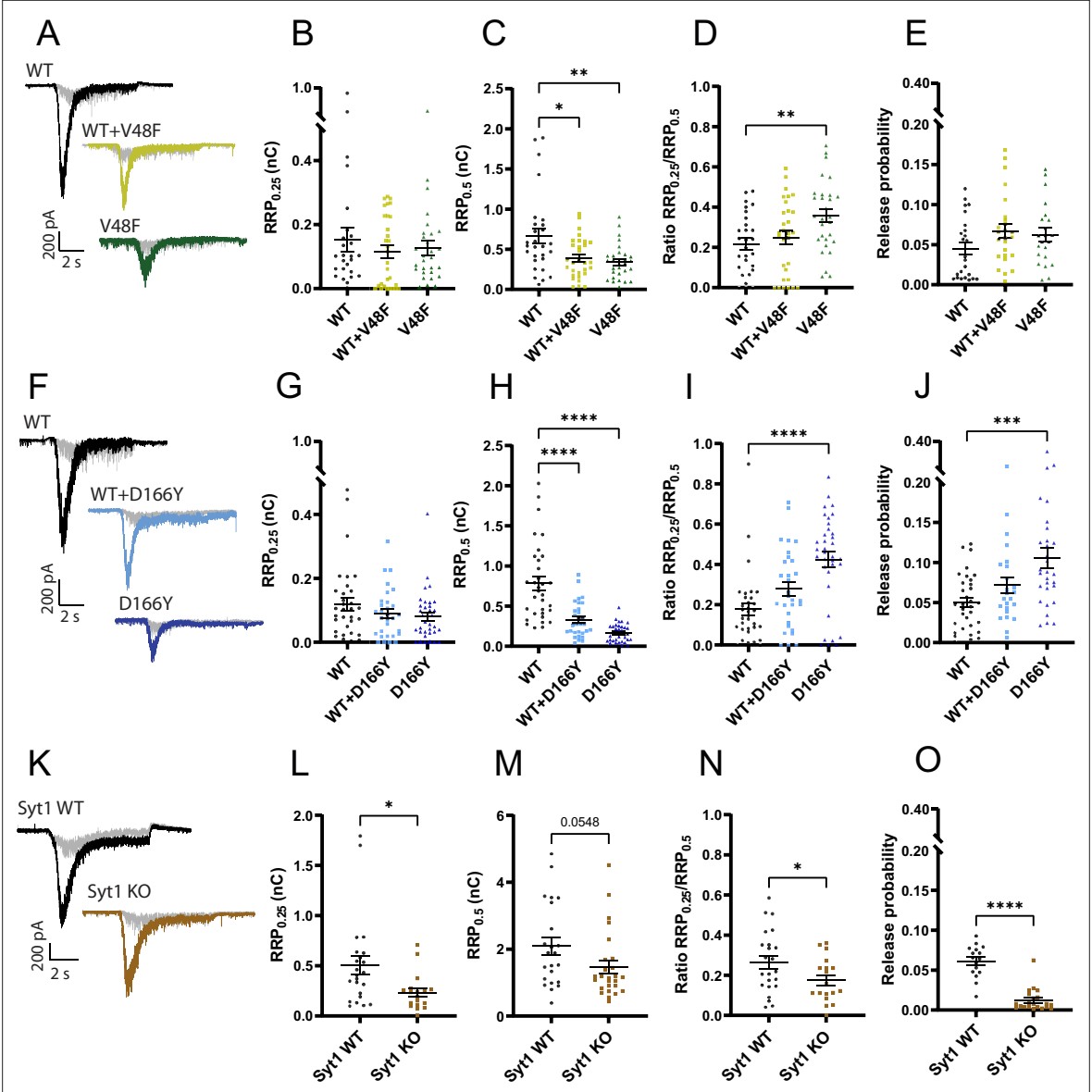

**Figure 5.** The apparent energy barrier for vesicle fusion is lowered by V48F and D166Y, but not by removing Syt1. (**A, F, K**) Example traces for the wildtype (WT), mutant, and co-expressed condition. Each cell was stimulated by 0.25 M (in gray) and 0.5 M sucrose (in black or color). (**B, G, L**) The charge released by 0.25 M sucrose (V48F: $n$ = 28, 30, 29 for WT, co-expressed, and mutant conditions, respectively; D166Y: $n$ = 33, 30, 35; Syt1: $n$ = 23, 18 for WT and knockout [KO]). Syt1: $p < 0.05$, Welch's $t$-test. (**C, H, M**) The charge released by 0.5 M sucrose (V48F: $n$ = 28, 30, 29 for WT, co-expressed, and mutant conditions, respectively; D166Y: $n$ = 33, 30, 35; Syt1: $n$ = 23, 26 for WT and KO). SNAP25b: ****$p < 0.0001$, **$p < 0.01$, *$p < 0.05$, Brown–Forsythe analysis of variance (ANOVA) with Dunnett's multiple comparisons test; Syt1: $p$ = 0.0548, unpaired $t$-test. (**D, I, N**) The ratio of 0.25 and 0.5 M sucrose pool (V48F: $n$ = 28, 30, 29 for WT, co-expressed, and mutant conditions, respectively; D166Y: $n$ = 33, 30, 35; Syt1: $n$ = 23, 18 for WT and KO). SNAP25b: ****$p < 0.0001$, **$p < 0.01$, ANOVA with Dunnett's multiple comparisons test; Syt1: *$p < 0.05$, unpaired $t$-test. (**E, J, O**) Release probability calculated by dividing the charge of an eEPSC with the 0.5 M sucrose pool (V48F: $n$ = 24, 25, 22 for WT, co-expressed, and mutant conditions, respectively; D166Y: $n$ = 33, 24, 30; Syt1, $n$ = 16, 21 for WT and KO). SNAP25b: ***$p < 0.001$, ANOVA with Dunnett's multiple comparisons test; Syt1: ****$p < 0.0001$, unpaired $t$-test.

The online version of this article includes the following source data for figure 5:

**Source data 1.** Quantitative data.

dissect the RRP with a short (few s) sucrose application. We set out to understand whether the V48F and D166Y mutants changed the size of the RRP.

Application of 0.5 M sucrose to neurons with the V48F or D166Y mutation resulted in estimates of the RRP (denoted $RRP_{0.5}$) that were significantly reduced compared to the WT condition (***Figure 5A,***

*C, F, H*). Also the co-expressed condition displayed significantly reduced $RRP_{0.5}$. Application of two different sucrose concentrations is used to probe the size of the energy barrier for release (*Basu et al., 2007*; *Schotten et al., 2015*), because the use of a lower sucrose concentration (typically 0.25 M) will only release a fraction of the RRP, and this fraction depends sensitively on the energy barrier for release; the size of the RRP at 0.5 M sucrose ($RRP_{0.5}$) is used for normalization. Application of 0.25 M sucrose to V48F and D166Y expressing neurons strikingly led to unchanged pool release ($RRP_{0.25}$) (*Figure 5B, G*). Consequently, the ratio of pools ($RRP_{0.25}/RRP_{0.5}$) was significantly increased for both the V48F and the D166Y mutations (*Figure 5D, I*), indicating that these two mutations decrease the apparent energy barrier for fusion (*Schotten et al., 2015*). The co-expressed conditions were in-between WT and mutant and did not reach statistical significance. Consistent with these results, both mutations on average increased the release probability, calculated as the ratio between the eEPSC charge and the $RRP_{0.5}$ charge measured in the same cell. The increase was statistically significant for the D166Y mutation (*Figure 5J*), but not for V48F (*Figure 5E*). This is different from the situation in the *Syt1* KO, where the $RRP_{0.25}/RRP_{0.5}$ and the release probability were both significantly decreased (*Figure 5K–O*), although in this dataset the reduction in $RRP_{0.5}$ size by removing Syt1 did not reach statistical significance (p = 0.0548, unpaired *t*-test). Thus, the increased mEPSC release rate from the *Syt1* KO does not correlate with a reduced energy barrier as assayed by sucrose, which was reported before (*Huson et al., 2020*), and the V48F and D166Y have specific gain-of-function features that lower the energy barrier for vesicle fusion.

To investigate the reasons for the change in RRP size by D166Y and V48F, we considered a one-pool model for the RRP (*Figure 6A*), where the RRP is filled by priming ($k_1$) from an upstream pool and depleted by depriming ($k_{-1}$) or spontaneous fusion ($k_f$). Dividing the miniature release rates ($r_{mini}$) with the size of the RRP yields the spontaneous fusion rate $k_f$. The current plateau during 0.5 M sucrose application (*Figure 6B*) essentially reports on the priming rate ($k_1$), providing that the fusion rate is sufficiently increased by sucrose (Materials and methods). However, high sucrose concentrations can change the baseline current level, which will cause an error in estimation of $k_1$. We therefore corrected the plateau level using a plot of the variance versus mean during the sucrose application (*Figure 6C*); this plot is linear for the type of noise generated by synaptic transmission (shot noise). Back-extrapolation of a regression line allows a determination of the baseline current level in the presence of sucrose (*Figure 6B, C*; see Materials and methods). Combining RRP size with the estimates of $k_1$ and $k_f$ allows determining the depriming rate, $k_{-1}$.

These calculations showed that $k_f$, the spontaneous fusion rate was strongly increased in the V48F and D166Y, and this increase was much larger for the mutations than for the *Syt1* KO (*Figure 6D*) fulfilling D166Y > V48F > *Syt1* KO. Further analysis showed that in both mutations, the forward priming rate, $k_1$, and the depriming rate, $k_{-1}$, were both decreased – the latter effect was only significant for D166Y (*Table 1*). Summing up the effects of changes in the three parameters on the RRP size for the V48F, D166Y, and *Syt1* KO (*Figure 6E*), we can conclude that for the V48F and even more for the D166Y, spontaneous release contributes to the reduction in RRP size, whereas this effect is minimal in the *Syt1* KO. However, the major reason for the smaller RRP size is a reduction in priming rate, $k_1$, which is partly counteracted by the decrease in $k_{-1}$ (which would increase RRP size). Overall, changes in priming, depriming, and spontaneous fusion rates combine to change RRP size.

Repetitive stimulation to determine the RRPs often results in lower estimates than sucrose application, because action potentials draw on a sub-pool of the RRP (*Moulder and Mennerick, 2005*), whereas sucrose releases the entire RRP (*Rosenmund and Stevens, 1996*). To determine the RRP sub-pool that evoked release draws on (denoted $RRP_{ev}$) we applied repetitive stimulation (50 APs @ 40 Hz) and used back-extrapolation to determine the RRP (*Neher, 2015*). Performing this in our standard 2 mM $Ca^{2+}$-containing extracellular solution resulted in overall smaller estimates for $RRP_{ev}$ for V48F and D166Y compared to WT (*Figure 7—figure supplement 1*); however, the differences were not statistically significant. The back-extrapolation method works best with high release probabilities (*Neher, 2015*); therefore, we repeated these experiments in the presence of 4 mM extracellular $Ca^{2+}$. Under these conditions, the $RRP_{ev}$ was significantly reduced for both the V48F and D166Y mutation (*Figure 7C, G*; *Figure 7—figure supplement 2*). Strikingly, the release probabilities (calculated as the charge of the first eEPSC of the train divided by the $RRP_{ev}$) were decreased for both mutations (*Figure 7D, H*), which correlated with an increased paired-pulse ratio (*Figure 7A, E*, *inserts*) (*Zucker and Regehr, 2002*). Thus, although the release probability calculated by normalizing evoked charge to

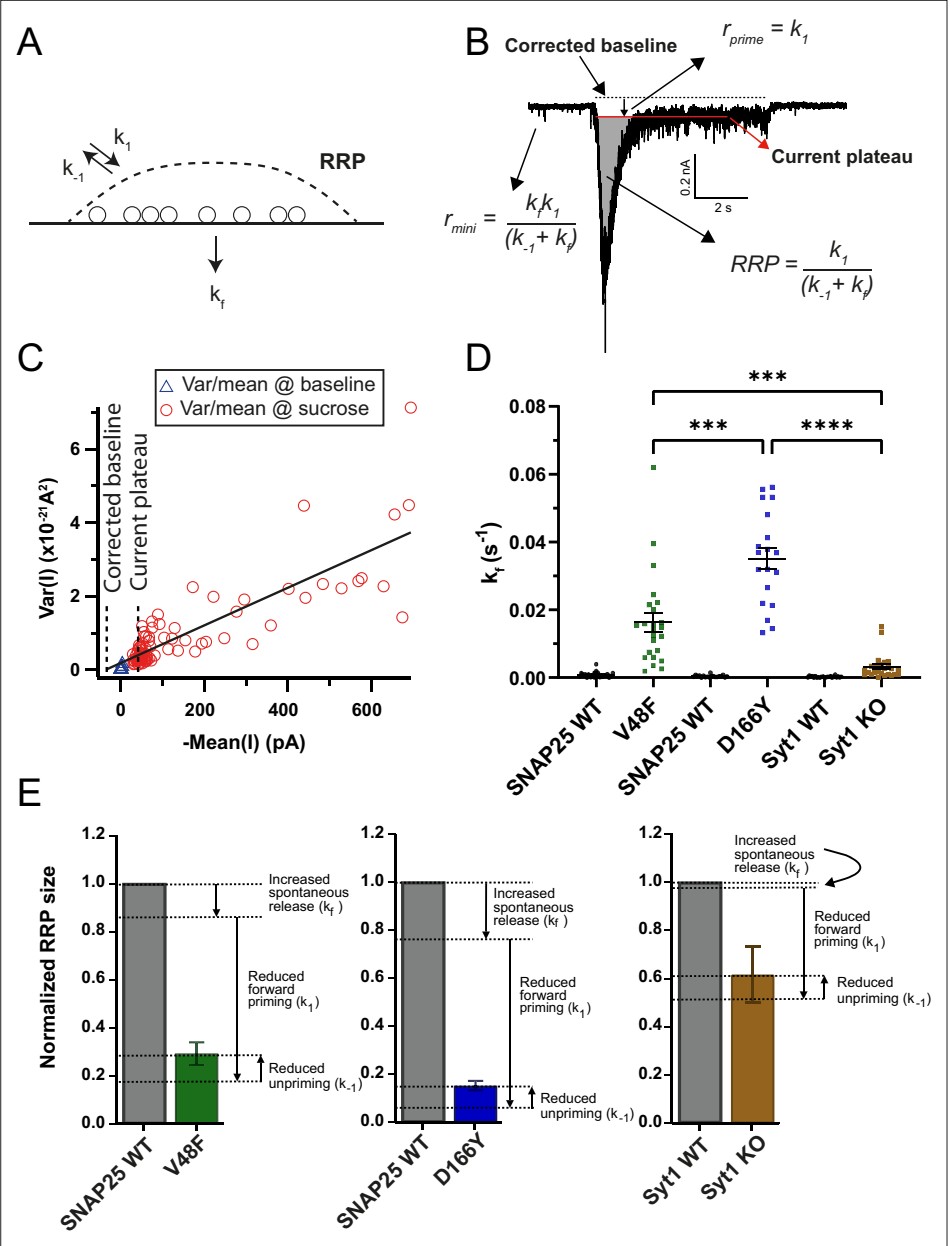

**Figure 6.** Dissection of the readily releasable pool (RRP) reduction in V48F and D166Y mutations. (**A**) One-pool model of the RRP. $k_1$ is the rate of priming (units vesicles/s), $k_{-1}$ is the rate of depriming ($s^{-1}$), and $k_f$ is the rate of fusion ($s^{-1}$). (**B**) Estimation of the three parameters from the response to 0.5 M sucrose and a measurement of the spontaneous release rate. (**C**) Variance-mean analysis in 50-ms intervals during the sucrose application allows determination of the corrected baseline by back-extrapolation of a regression line to the variance of the baseline. (**D**) Normalized mEPSC frequency ($k_f$) for V48F, D166Y, and Syt1 knockout (KO) (V48F: $n = 23$, 24 for wildtype [WT] and mutant conditions, respectively; D166Y: $n = 19$, 19; Syt1: $n = 23$, 26). Brown–Forsythe analysis of variance (ANOVA) test with Dunnett's multiple comparison test, testing the three mutant conditions against each other. ****$p < 0.0001$, ***$p < 0.001$. (**E**) Normalized RRP size for WT and mutant conditions, with indications of the effect of the mutant-induced changes in $k_1$, $k_{-1}$, and $k_f$ on the RRP size.

The online version of this article includes the following source data and figure supplement(s) for figure 6:

**Source data 1.** Excel file containing quantitative data.

**Figure supplement 1.** Effect of sucrose stimulation on estimates of $k_1$ and $k_{-1}$.

**Table 1.** Estimated parameters affecting the size of the readily releasable pool (RRP).

Displayed is mean ± standard error of the mean (SEM). Two-sample $t$-test or Welch's $t$-test comparing mutant to wildtype (WT): *p < 0.05; ***p < 0.001; ****p < 0.0001, #non-significant (p = 0.125), °non-significant (p = 0.210).

| Mean ± SEM | WT | V48F | WT | D166Y | *Syt1* WT | *Syt1* KO |
|---|---|---|---|---|---|---|
| $k_1$ [vesicles/s] | 385.6 ±52.2 | 79.87*** ±12.5 | 457.4 ±77.4 | 37.68**** ±7.33 | 1227 ±189 | 646* ±114 |
| $k_{-1}$ [1/s] | 0.0903 ±0.0091 | 0.0605# ±0.011 | 0.1114 ±0.012 | 0.0294**** ±0.0122 | 0.140 ±0.016 | 0.114° ±0.013 |
| $k_f$ [1/s] | 0.000844 ±0.000178 | 0.0164**** ±0.00230 | 0.000398 ±0.000077 | 0.03522**** ±0.00352 | 0.000235 ±0.000043 | 0.00286*** ±0.00062 |

the sucrose-determined RRP (RRP$_{0.5}$) was increased (see above), the release probability when normalizing to RRP$_{ev}$ was decreased. This points to a difference in the organization of the RRP$_{ev}$ and RRP$_{0.5}$ (see Discussion). The forward priming rate is determined as the slope of the linear fit used for the back-extrapolation; this parameter was reduced in both mutations (*Figure 7B, F*; *Figure 7—figure supplement 2*). Thus, the difference in priming rate extends to the RRP$_{ev}$.

Overall, these data demonstrate a rather complex phenotype of the D166Y and V48F mutations, which combine a lowering of the energy barrier – a gain-of-function feature – with a loss of vesicle priming – a loss-of-function feature. The D166Y and V48F phenotypes can be summarized as (1) desynchronized eEPSCs with at most mildly reduced total charge, (2) lowered energy barrier for fusion, (3) increased release probability when normalized to the sucrose pools, (4) decreased RRP size due to unclamped spontaneous release and lowered forward priming rates, and (5) short-term facilitation. These phenotypes are distinctive from the Syt1 KO, which does not have a preserved charge of the eEPSC, or lowered energy barrier when probed by sucrose, or an increased release probability. Thus, the D166Y and V48F cannot be understood solely in terms of a lack of Syt1 coupling; instead, gain-of-function features are present in the mutants which are absent upon deletion of Syt1.

## V48F and D166Y mutants show increased partner SNARE interactions and cause unregulated fusion in vitro

We next tried to identify the biochemical properties of V48F and D166Y, which could support a gain-of-function phenotype during exocytosis. These data are displayed in *Figures 8 and 9* – data on the I67N mutant were obtained in parallel and will be presented later. To test to which degree the mutants may change the stability of SNARE complexes, full-length t-SNAREs (syntaxin-1 and SNAP25) were incubated with the cytosolic domain of VAMP2 (VAMP2cd) overnight. Cis-SNARE complex stability was tested in the presence of SDS at the indicated temperatures (*Figure 8A*), and the release of syntaxin-1 as a single-protein band was used as a measure of the complex dissociation. The WT, V48F, and D166Y v-/t-SNARE complexes showed a similar stability with half-maximal dissociation occurring at approximately 71°C.

Next, we asked whether the V48F and D166Y mutants would change interaction with Syt1. To this end, Atto647-labeled giant unilamellar vesicles (GUVs) filled with isosmotic sucrose and containing preassembled t-SNARE complexes were preincubated with Atto488/Atto550-labeled small unilamellar vesicles (SUVs) containing Syt1 as well as VAMP2 for 10 min on ice, followed by centrifugation to re-isolate GUVs with attached SUVs. Fusion was blocked by performing the assay on ice. Attachment of SUVs was determined by measuring the Atto550 fluorescence of SUVs. In the absence of PI(4,5)P$_2$, vesicle attachment occurs by Syt1:SNARE interactions (*Kim et al., 2012*; *Parisotto et al., 2012*), probably involving the primary interface (*Zhou et al., 2015*). In the cell, PI(4,5)P$_2$ binding by Syt1 appears to happen first (*Honigmann et al., 2013*), whereas subsequent Syt1:SNARE binding leads to a tightly docked state (*Chen et al., 2021*). Under our conditions, both SNAP25 mutants (V48F and D166Y) showed significantly impaired attachment of Syt1/VAMP2 SUVs to t-SNARE GUVs (*Figure 8B*). Both V48 and D166 directly interact with Syt1 (*Figure 1B, C*, *Zhou et al., 2015*), so that changing these two amino acids to more bulky or hydrophobic amino acids reduced the docking from 42.7 ± 2.6% WT docking efficiency to 29.6 ± 4.1% and 20.9 ± 3.7%, respectively. In the presence of 1% PI(4,5)P$_2$, the vesicle attachment increased from 42.7 ± 2.6% to 66.0 ± 0.5% for WT t-SNARE. Although the Syt1:PI(4,5)P$_2$ interaction predominated SUV–GUV docking, both mutants still showed

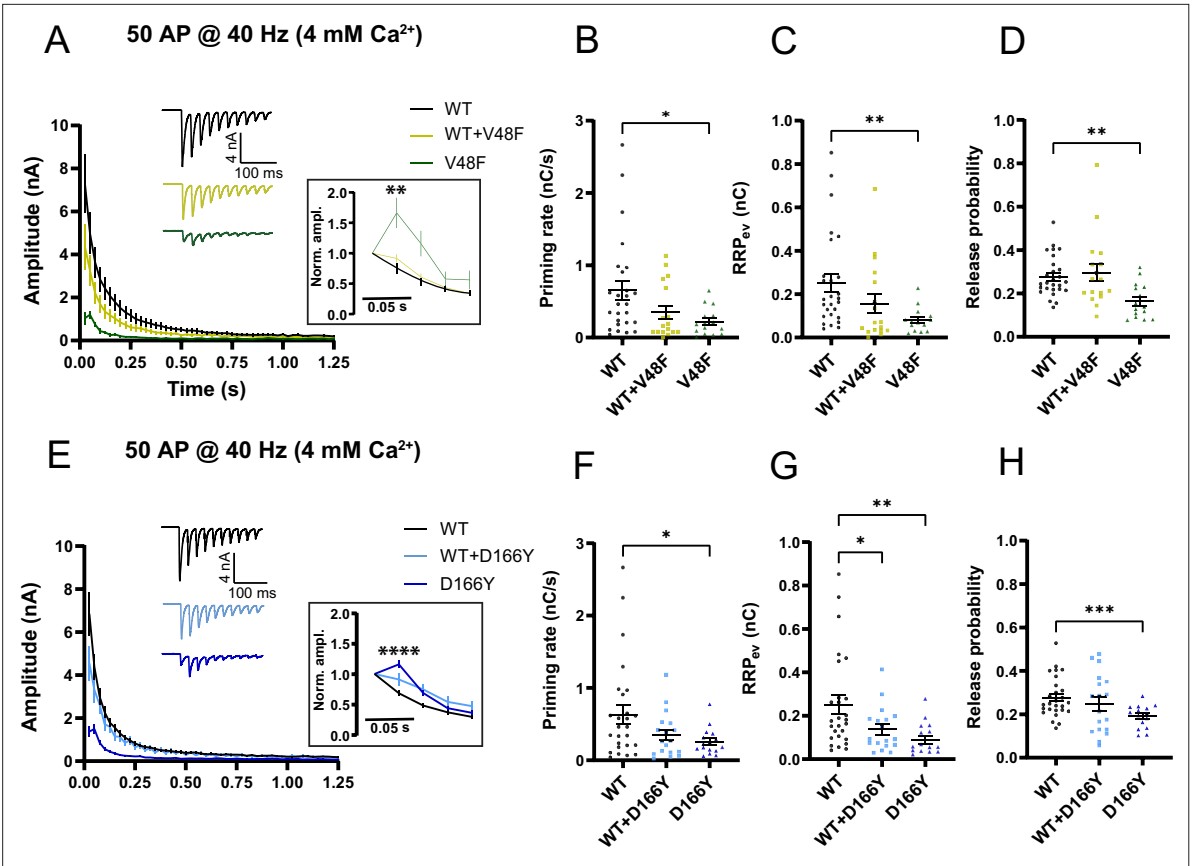

**Figure 7.** SNAP25 V48F and D166Y mutations change short-term plasticity toward facilitation. (**A, E**) eEPSCs in response to 50 APs at 40 Hz recorded in 4 mM extracellular Ca$^{2+}$ (V48F: 27, 17, 15 for wildtype [WT], co-expressed, and mutant conditions, respectively; D166Y: 27, 18, 16). Inserts: Normalized eEPSC amplitudes demonstrating facilitation of mutant conditions. ****p < 0.0001; **p < 0.01, Brown–Forsythe analysis of variance (ANOVA) with Dunnett's multiple comparison test. (**B, F**) Priming rate calculated as the slope of a linear fit to the cumulative evoked charges during the last part of stimulation (V48F: 27, 17, 15 for WT, co-expressed, and mutant conditions, respectively; D166Y: 27, 18, 16). *p < 0.05, ANOVA (V48F) or Brown–Forsythe ANOVA (D166Y) with Dunnett's multiple comparisons test. (**C, G**) Readily releasable pool (RRP) calculated by back-extrapolation of a linear fit to the cumulative evoked charges during the last part of stimulation (V48F: 27, 17, 15 for WT, co-expressed, and mutant conditions, respectively; D166Y: 27, 18, 16). **p < 0.01, *p < 0.05, ANOVA (V48F), or Brown–Forsythe ANOVA (D166Y) with Dunnett's multiple comparisons test. (**D, H**) Release probability calculated as the charge of the first evoked response divided by the RRP obtained by back-extrapolation (V48F: 27, 17, 15 for WT, co-expressed, and mutant conditions, respectively; D166Y: 27, 18, 16). ***p < 0.001; **p < 0.01, ANOVA (V48F), or Brown–Forsythe ANOVA (D166Y) with Dunnett's multiple comparisons test.

The online version of this article includes the following source data and figure supplement(s) for figure 7:

**Source data 1.** Excel file containing quantitative data.

**Figure supplement 1.** Train stimulations of V48F and D166Y in 2 mM Ca$^{2+}$.

**Figure supplement 1—source data 1.** Excel file containing quantitative data.

**Figure supplement 2.** Cumulative charges of V48F and D166Y trains in 4 mM Ca$^{2+}$.

---

significantly reduced vesicle attachment by approximately 6% compared to WT (***Figure 8C***) (V48F: 59.1 ± 1.4% and D166Y: 60.0 ± 2.0%).

To understand how the mutations affect fusion in a well-defined reconstituted membrane fusion system, we performed in vitro lipid mixing assays using GUVs containing both t-SNAREs (syntaxin-1 and SNAP25) with 1% PI(4,5)P$_2$, and 0.5% Atto488/0.5% Atto550-labeled SUVs containing Syt1 and VAMP2 in the presence or absence of complexin-II (6 μM) (***Kedar et al., 2015***; ***Malsam et al., 2012***). Fusion was measured at 37°C by Atto488 fluorescence dequenching, which occurs upon lipid mixing with GUVs. Calcium (100 μM free Ca$^{2+}$ final) was added after 2 min to the t-SNARE GUV assay. Measurements were continued for another 5 min. SUVs treated with botulinum toxin D, which cleaves VAMP2 and abolishes membrane fusion, served as negative control and the corresponding

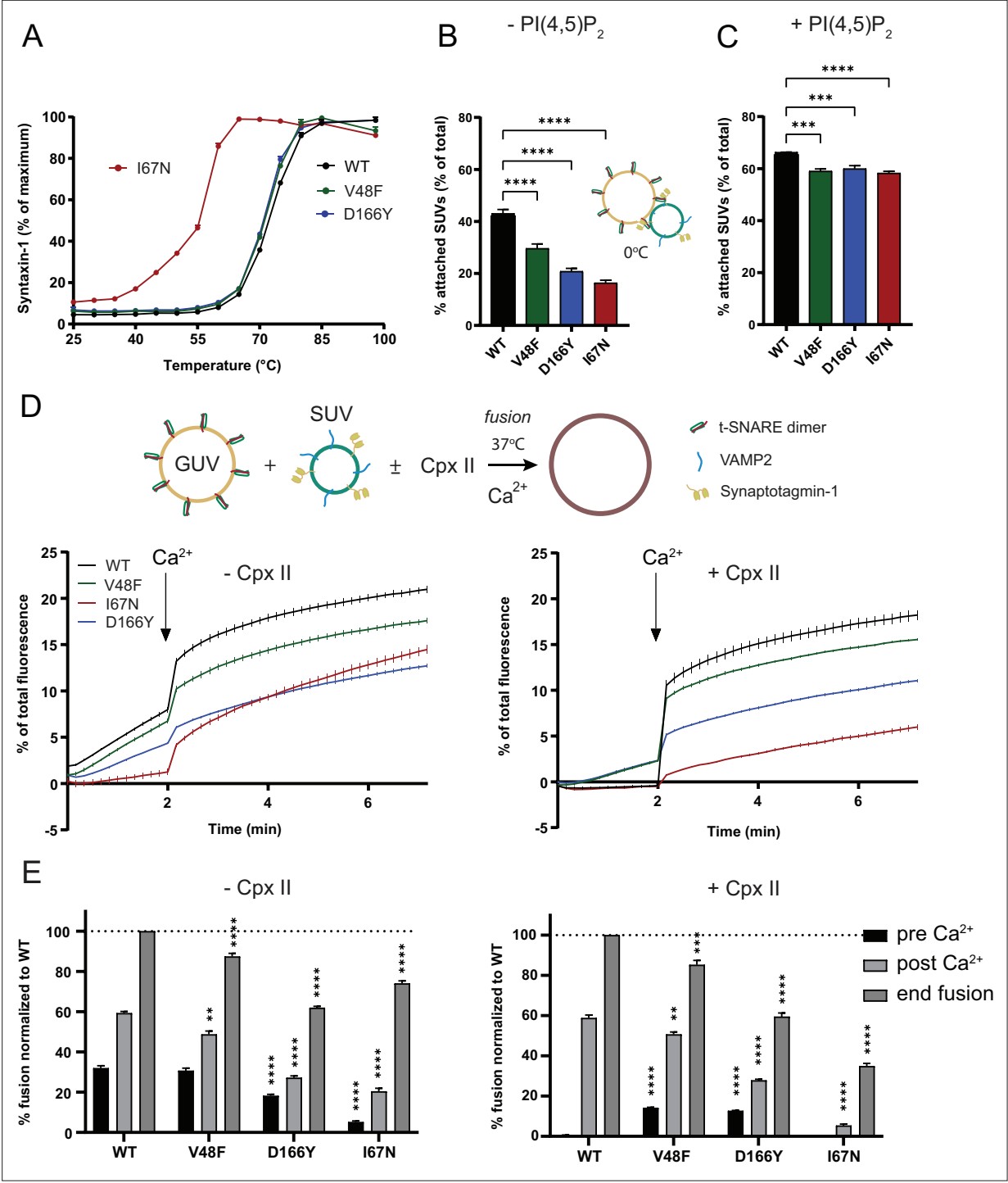

**Figure 8.** Pathogenic SNAP25 mutations affect synaptotagmin-1 interaction and fusion rates in vitro. (**A**) In the presence of SDS, SNAP25b I67N containing v-/t-SNARE complexes were more sensitive to temperature-dependent dissociation. Shown are mean ± standard error of the mean (SEM; $n = 3$) for SNARE complexes including SNAP25b wildtype (WT) and the I67N, V48F, and D166Y mutations. (**B**, **C**) In vitro Syt1/VAMP2 small unilamellar vesicles (SUVs) docking to t-SNARE giant unilamellar vesicles (GUVs) was significantly reduced by SNAP25b V48F, I67N, and D166Y mutants either in absence (**B**) or presence (**C**) of PI(4,5)P$_2$. Fusion was blocked by performing the assay on ice. ****$p < 0.0001$; ***$p < 0.001$, analysis of variance (ANOVA) with Dunnett's multiple comparison test. (**D, E**) In vitro lipid mixing assays of VAMP/Syt1 SUVs with t-SNARE GUVs containing SNAP25b V48F, I67N, or D166Y mutants showed impaired membrane fusion in the absence (left) or presence (right) of complexin-II. Fusion clamping in the presence of complexin was selectively reduced by V48F and D166Y. Bar diagrams show lipid mixing just before (pre) and after (post) Ca$^{2+}$ addition and at the end of

*Figure 8 continued on next page*

*Figure 8 continued*

the reaction. Shown is mean ± SEM (*n* = 3). ****p < 0.0001; **p < 0.01, ANOVA with Dunnett's multiple comparisons test, comparing each mutation to the corresponding WT condition.

The online version of this article includes the following source data for figure 8:

**Source data 1.** Excel file containing quantitative data.

background fluorescence was subtracted. Measurements were normalized to total fluorescence after detergent lysis.

Complexin-II clamps spontaneous $Ca^{2+}$-independent membrane fusion in the reconstituted assay (i.e. fusion before addition of $Ca^{2+}$), via laterally binding the membrane-proximal C-terminal ends of SNAP25 and VAMP2 (*Malsam et al., 2020*). Remarkably, V48F and D166Y showed impaired clamping by complexin, as apparent by increased fusion before $Ca^{2+}$ addition (*Figure 8D*). The decreased clamping is likely caused by the reduced interaction of V48F and D166Y with Syt1. As a note, the clamping function of Syt1 becomes only obvious in the presence of complexin, whereas the clamping function of complexin depends on the presence of Syt1 (*Malsam et al., 2012*); thus, the clamping function of the two proteins cannot be separated. After $Ca^{2+}$ triggering, WT SNAP25 supported the largest amount of fusion, followed by V48F, whereas D166Y $Ca^{2+}$-dependent fusion was clearly reduced (*Figure 8D*). Notably, this is the same sequence as found for evoked release in synapses, considering either the amplitude or the charge of the eEPSC (*Figure 4B, C, F, G*). Thus, the in vitro assay reproduces both the increased spontaneous release and the reduced $Ca^{2+}$-dependent release found in neurons, indicating that these features are present within the minimal set of fusion proteins included in this assay (i.e. the SNAREs, Syt1 and complexin).

There is evidence that SNAP25 might enter the SNARE complex last, after syntaxin-1 and VAMP2 are joined by Munc18-1 (*Baker et al., 2015*; *Jiao et al., 2018*; *Sitarska et al., 2017*), although another view is that a syntaxin/SNAP25 dimer bound to Munc18-1 acts as an intermediary (*Jakhanwal et al., 2017*). In the former case, mutations changing association of SNAP25 to the SNAREs might change exocytosis efficiency. To test this, we used a syntaxin-1 GUV assay, where the incorporation of soluble SNAP25 into the SNARE complex becomes a rate-limiting step. In this assay, fusion kinetics are much slower when compared with fusion reactions containing GUVs with preassembled t-SNAREs. Accordingly, we allowed 30 min for pre-stimulation fusion to take place, and after addition of $Ca^{2+}$ (100 µM free $Ca^{2+}$), fusion was followed for another 30 min (*Figure 9A*). The assay was performed in the presence and absence of complexin-II. D166Y and to a lesser degree V48F revealed enhanced stimulation of fusion in comparison to WT before calcium was added, regardless of the presence or absence of complexin (*Figure 9B*). Comparing data with and without complexin established that complexin barely suppressed spontaneous fusion in reactions containing the V48F mutant. The presence of complexin partially reduced $Ca^{2+}$-independent fusion, but D166Y still stimulated pre-$Ca^{2+}$ fusion compared to WT. Overall, these data demonstrate that D166Y and V48F are gain-of-function mutants under conditions where SNAP25 association to the other SNAREs is rate limiting.

To test directly whether V48F and especially D166Y enhance SNARE interactions, co-flotation assays were performed. SUVs containing either syntaxin-1 (Stx-1), or VAMP2, or Syt1, or the Syt1/VAMP2 combination were incubated with SNAP25 and re-isolated by flotation using a Nycodenz density gradient. An additional reaction, reflecting the fusion assay, contained SNAP25 in combination with both Syt1/VAMP2 SUVs and Syntaxin-1 SUVs (*Figure 9D*). SNAP25 recruitment for each condition was determined by SDS–PAGE followed by Coomassie Blue and silver staining. Silver stained SNAP25 bands were quantified, and the mutants were plotted relatively to the WT SNAP25 (*Figure 9D* and *Figure 9—figure supplement 1*).

SNAP25 WT did not show any binding to Syt1 SUVs (*Figure 9—figure supplement 1*), although direct interactions with Syt1 would be predicted based on the primary interface, indicating that such interactions are not stable under the employed conditions, which is expected because SNAP25 is unstructured until it binds its SNARE partners (*Fasshauer et al., 1997*). D166Y showed profoundly increased interactions with SUVs containing either syntaxin-1, or VAMP2, or Syt1/VAMP2, and the combination used in the fusion assay (*Figure 9D*; *Figure 9—figure supplement 1*). V48F displayed mildly increased binding to syntaxin-1 and Syt1/VAMP1 SUVs, and a tendency to increased association to VAMP2 SUVs (p = 0.0629, one-sample *t*-test). These data show that loss of Syt1 interaction upon mutation in the primary interface can be accompanied by a gain-of-function phenotype stimulating

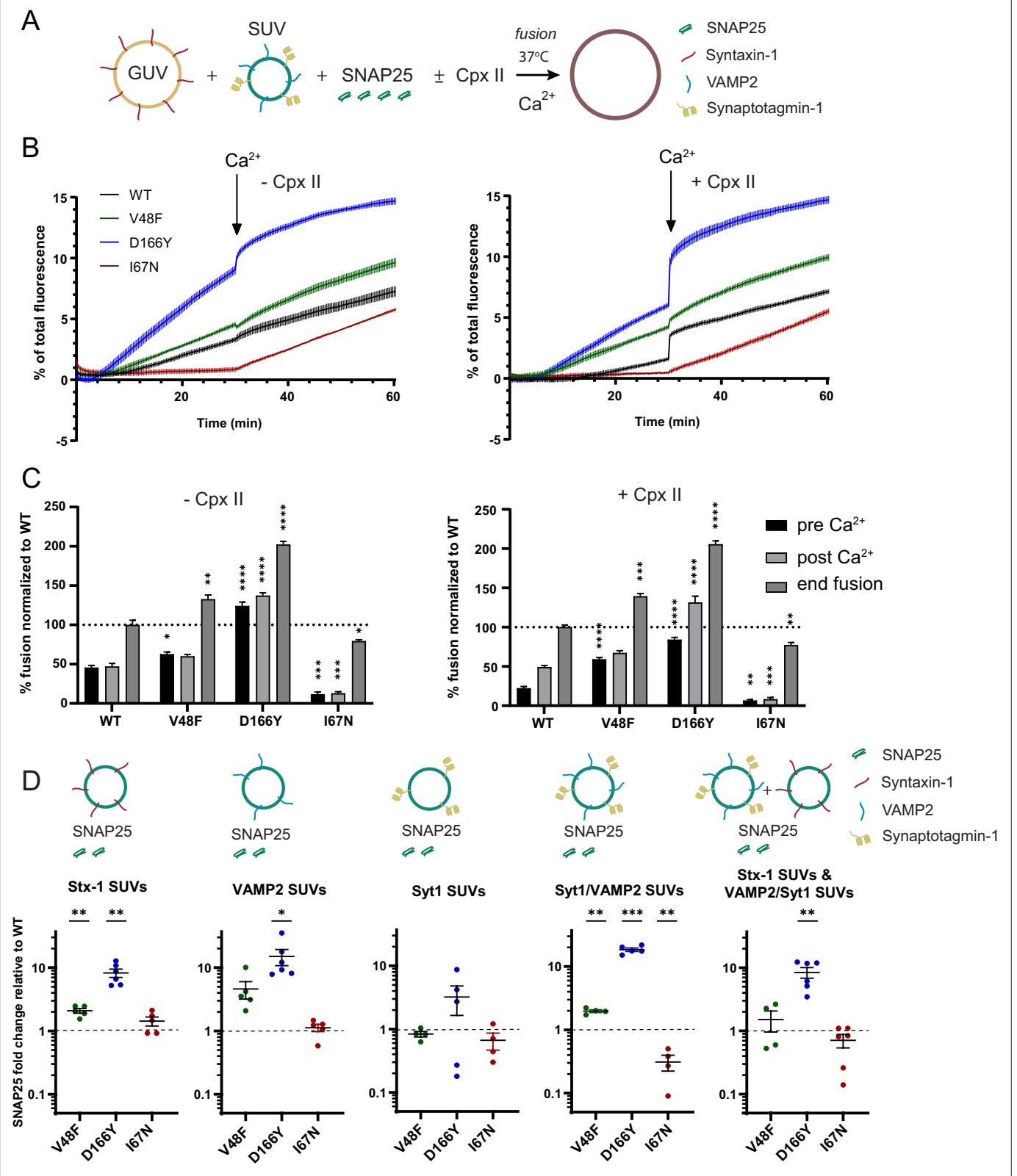

**Figure 9.** The D166Y mutation increases binding to its SNARE partners. (**A–C**) In vitro lipid mixing assays of VAMP/Syt1 small unilamellar vesicles (SUVs) with syntaxin-1A giant unilamellar vesicles (GUVs) in the presence of soluble SNAP25b. V48F and D166Y mutants showed impaired fusion clamping in the absence (left) or presence (right) of complexin-II; I67N (red) showed impaired $Ca^{2+}$-independent and $Ca^{2+}$-triggered fusion. Bar diagrams show lipid mixing just before (pre) and after (post) $Ca^{2+}$ addition and at the end of the reaction. Mean ± standard error of the mean (SEM; $n$ = 3). ****p <

*Figure 9 continued on next page*

Figure 9 continued

0.0001; ***p < 0.001; **p < 0.01; *p < 0.05, analysis of variance (ANOVA) with Dunnett's multiple comparisons test, comparing each mutation to the corresponding wildtype (WT) condition. (**D**) SNAP25b D166Y showed enhanced interactions with SUVs carrying reconstituted syntaxin-1A (Stx-1), VAMP2, Syt1/VAMP2, or an SUV mixture containing Syntaxin-1A and VAMP2/Syt1 in co-flotation assays, whereas V48F displayed weaker increases in interactions with SUVs containing syntaxin-1A, or Syt1/VAMP2. Shown is mean ± SEM on a logarithmic scale. ***p < 0.001, **p < 0.01, *p < 0.05, two-tailed one-sample *t*-test comparing to 1.

The online version of this article includes the following source data and figure supplement(s) for figure 9:

**Source data 1.** Excel file containing quantitative data.

**Figure supplement 1.** Floatation assay.

**Figure supplement 1—source data 1.** Original files for the analysis by Coomassie and silver stained gels.

**Figure supplement 1—source data 2.** PDF containing original pictures of gels with highlighted bands and sample labels.

**Figure supplement 2.** Molecular dynamics simulations of mutants.

**Figure supplement 2—source data 1.** Excel file with quantitative data.

interactions with the other SNARE partners. This association of SNAP25 to the other SNAREs might happen as one of the last steps toward fusion; consequently, when D166Y and V48F join the complex prematurely, it will bypass layers of control and result in uncontrolled fusion.

## The I67N mutation supports an intact RRP, but an increased energy barrier for fusion

We next addressed the phenotype of the I67N disease mutation, which was found in an 11-year-old female, who suffered from myasthenia, cortical hyperexcitability, ataxia, and intellectual disability, but with normal brain MRI (**Shen et al., 2014**). The I67 is found within the interaction layer +4 (**Fasshauer et al., 1998**), which helps in assembly of the C-terminal of the SNARE complex (**Gao et al., 2012**), and it might therefore have a different synaptic phenotype than V48F and D166Y.

In in vitro experiments, SNARE complexes formed with the I67N mutant displayed a lower stability, with the melting temperature reduced from 71°C to approximately 56°C (**Figure 8A**), as expected for a mutation that destabilizes the SNARE complex. I67N also showed a strong decrease in SUV docking (**Figure 8B, C**), which is likely caused by the destabilization of the t-SNARE complex, which indirectly perturbs the Syt1-binding interface(s). In lipid mixing assays, I67N strongly reduced both $Ca^{2+}$-independent and $Ca^{2+}$-dependent fusion, whether the t-SNAREs were preassembled (**Figure 8D, E**) or not (**Figure 9B, C**). The co-flotation assay did not display any binding of I67N to t-SNAREs or Syt1 (**Figure 9D**). The binding to Syt1/VAMP2 SUVs was reduced compared to SNAP25 WT, but since binding of the WT protein is already very low, the biological significance of this result is unclear. Overall, these data indicate that I67N is inferior in membrane fusion.

Lentiviruses encoding the I67N mutant N-terminally fused to EGFP expressed similar amounts as WT EGFP-SNAP25b (**Figure 10A–C**). Expression in SNAP25 KO neurons resulted in reduced rescue of survival in neurons expressing I67N alone, whereas neurons co-expressing WT and I67N had intermediate survival, not significantly different from WT (**Figure 10D**). Staining against MAP2 (dendritic marker) and VGlut1 (synapse marker) showed that the number of synapses on average was reduced in the I67N (**Figure 10E, F**), and the dendritic length was on average reduced (**Figure 10G**), but the changes did not reach statistical significance. Patch-clamp measurements demonstrated strongly reduced spontaneous release frequencies (**Figure 10H–J**) and evoked release amplitude (**Figure 10K, L**) with the I67N mutation, see also **Alten et al., 2021**. mEPSCs were absent in most I67N expressing neurons, whereas WT and I67N co-expressing neurons had a very low mEPSC rate, much closer to the I67N than the WT phenotype; similar for eEPSC amplitudes and charges (**Figure 10L-M**). The I67N therefore is dominant negative for both types of release, in contrast to the incompletely dominant phenotypes of the V48F and D166Y mutants (see above). The fraction of synchronous release was unchanged in WT and I67N co-expressing neuron (**Figure 10N**); this number could not be estimated for I67N expressing neurons due to the low amount of release.

Reduced spontaneous and evoked release could result from a decrease in priming, or fusion, or both. To distinguish between these possibilities, we turned to sucrose applications. Application of 0.25 M sucrose did not lead to any measurable release in I67N expressing cells, and only minimal release in cells co-expressing WT and I67N (**Figure 11B**). This indicates that the energy barrier is

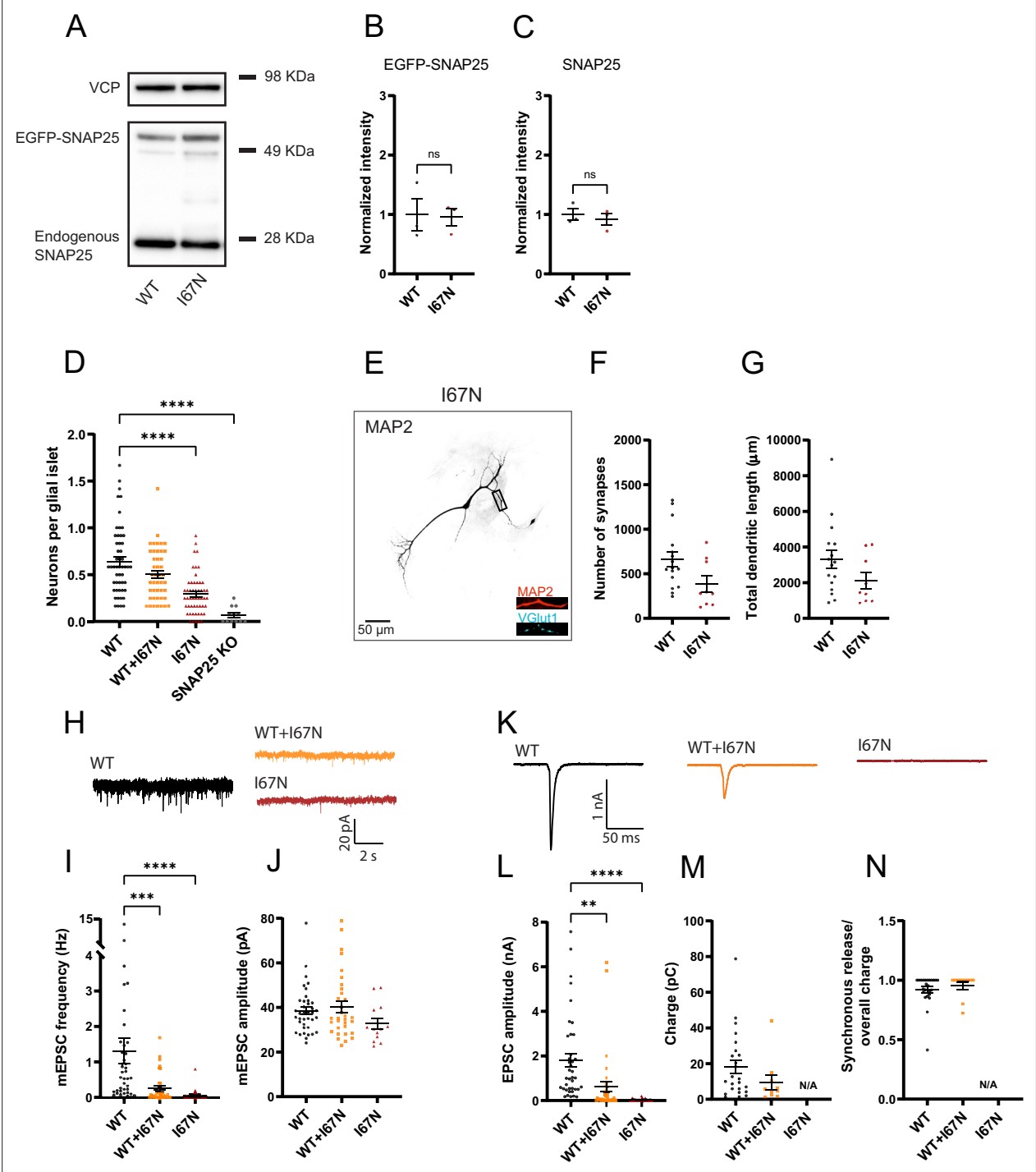

**Figure 10.** The I67N mutation inhibits spontaneous and evoked release. (**A**) SNAP25b I67N is similarly expressed as the wildtype (WT) SNAP25b protein. EGFP-SNAP25b was overexpressed in neurons from CD1 (WT) mice; both endogenous and overexpressed SNAP25 are shown. Valosin-containing protein (VCP) was used as the loading control. Quantification of EGFP-SNAP25b (**B**) and endogenous SNAP25 (**C**) from Western blots (as in A). Displayed are the intensity of EGFP-SNAP25b or endogenous SNAP25 bands, divided by the intensity of VCP bands, normalized to the WT situation (*n* = 3 independent experiments). The expression level of the I67N mutant was indistinguishable from WT protein (analysis of variance, ANOVA). (**D**) Cell viability represented as the number of neurons per glial islet. ****p < 0.0001, Brown–Forsythe ANOVA test with Dunnett's multiple comparisons test. (**E**) Representative image of mutant (I67N) hippocampal neurons stained for the dendritic marker MAP2 and the synaptic markers VGlut1. Displayed is MAP2 staining, representing the cell morphology, in inserts MAP2 staining is depicted in red and VGlut staining in cyan. The scale bar represents 50 μm. (**F**) Number of synapses per neuron in WT and mutant cells. The WT data are the same as in *Figure 2D, E* because these experiments were carried out in parallel. The difference was tested using ANOVA between all conditions, which was non-significant. (**G**) Total dendritic length of WT and

*Figure 10 continued*

mutant neurons. (**H**) Example traces of mEPSC release for WT, mutant (I67N), and 1:1 co-expression of WT and SNAP25 mutant. (**I**) The mini frequency was decreased in both I67N mutant and the WT + I67N combination (I67N: n = 39, 36, 30 for WT, co-expressed and mutant). ****p < 0.0001, ***p < 0.001, Kruskal–Wallis with Dunn's multiple comparisons. (**J**) mEPSC amplitudes were unchanged by the I67N mutation. (**K**) Example evoked excitatory postsynaptic currents (eEPSC) for WT, mutant (I67N), and co-expressed WT and mutant. (**L**) eEPSC amplitude was decreased by the I67N mutations (I67N: n = 39, 37, 30 for WT, co-expressed and mutant conditions, respectively). SNAP25b mutations: ****p < 0.0001, **p < 0.01, Brown–Forsythe ANOVA test with Dunnett's multiple comparisons test. (**M**) Overall evoked charge after a single depolarization (I67N: 24, 10, 0 for WT, co-expressed and mutant conditions, respectively). (**N**) Fractional contribution of the synchronous release component to the overall charge (I67N: 24, 10, 0 for WT, co-expressed, and mutant conditions, respectively). Source Data containing quantitative data are found in the Source Data files for *Figures 2–4*.

increased in amplitude, and therefore the RRP might be underestimated when probed by 0.5 M sucrose (*Schotten et al., 2015*). Indeed, 0.5 M sucrose displayed reduced release in the I67N and WT + I67N condition (*Figure 11C*), but this could be due to defects in priming or fusion. To investigate this, we applied a stronger stimulus, 0.75 M sucrose, to these cells (*Schotten et al., 2015*). Strikingly, the RRP as assessed by 0.75 M sucrose ($RRP_{0.75}$) was unchanged between WT, I67N, and WT + I67N co-expressed cells (*Figure 11E, G*). Application of 0.375 M sucrose led to small amounts of release in the I67N, but more in the WT + I67N co-expressed situation (*Figure 11F*). Forming the ratio $RRP_{0.375}$/$RRP_{0.75}$ revealed a statistically significant reduction in I67N compared to WT (*Figure 11H*). Therefore, the RRP per se appears intact in the I67N (when probed by sufficiently high concentrations of sucrose), but the vesicles face a higher apparent fusion barrier, which makes the RRP appear smaller if assessed by 0.5 M sucrose. The higher apparent fusion barrier explains the lower frequency of mEPSC in the I67N mutation, the lower degree of spontaneous fusion in in vitro assays, as well as the lower amount of $Ca^{2+}$-dependent release in vitro and in the cell.

The I67N mutation profoundly affected trains in 2 mM extracellular $Ca^{2+}$ (*Figure 11I*) leading to strong facilitation, which is expected due to the strong phenotype of this mutation, which radically lowers release probability. Even when co-expressed with WT protein, the train facilitated over the first several stimulations, attesting to the strong dominant-negative feature of the I67N mutation (*Figure 11I–J*). Consequently, the paired-pulse ratio was increased in the I67N and intermediate in the WT + I67N condition (*Figure 11K*). Back-extrapolation of these trains was not reliable, because the low release probability in the I67N mutation made it impossible to achieve sufficient depletion of the RRP.

The energy barrier for fusion is exquisitely sensitive to the charges on the surface of the SNARE complex, with positive charges decreasing and negative charges increasing the fusion barrier amplitude (*Ruiter et al., 2019*). To investigate whether the same electrostatic mechanism applies to the I67N, we combined the I67N mutation with a mutation of four amino acids ('4K' = SNAP25 E183K/S187K/T190K/E194K) in the second SNARE domain of SNAP25, constructing the quintuple mutation ('I67N/4K' = SNAP25 I67N/E183K/S187K/T190K/E194K). The 4K mutation lowers the energy barrier for fusion by increasing the charge of the SNARE complex surface by +6 via charge introduction and charge reversal (*Ruiter et al., 2019*). The 4K mutation increased the mEPSC release rate compared to WT (*Ruiter et al., 2019*), whereas in the combined I67N/4K mutation the spontaneous release rate was indistinguishable from WT (*Figure 12A–C*), showing that increased positive charges rescued the defect of spontaneous release in the I67N mutant. Evoked release was also increased in the 4K mutation compared to WT (*Ruiter et al., 2019*), but in the I67N/4K mutation, evoked release was still strongly depressed compared to WT (*Figure 12D, E*). Nevertheless, evoked release was noticeable in the combined mutation, whereas it was almost absent in I67N expressing cells (comp. *Figure 10K, L*), indicating a positive effect of the 4K mutation, which amounted to an increase by a factor ~5 (eEPSC amplitude, I67N: 0.0475 ± 0.0087; I67N/4K: 0.2668 ± 0.12 nA; Mann–Whitney test, p = 0.035).

We previously created a simple mathematical model that links the release rate to the number of charges added to the SNARE complex (*Ruiter et al., 2019*). This model includes both spontaneous and evoked release, which are separated by the addition of 35 positive charges in the latter case (*Figure 12F*, black points are WT spontaneous and evoked release rates; blue line is the model fitted to WT data). Placing the spontaneous release rates for I67N and I67N/4K on this curve (by dividing the spontaneous release rate with RRP size and finding a corresponding charge value using the model) resulted in two points (red) separated by 5.6 charges, which is close to the nominal 6 charges added by the 4K mutation (*Figure 12F*). Similarly, evoked release in the I67N and the I67N/4K

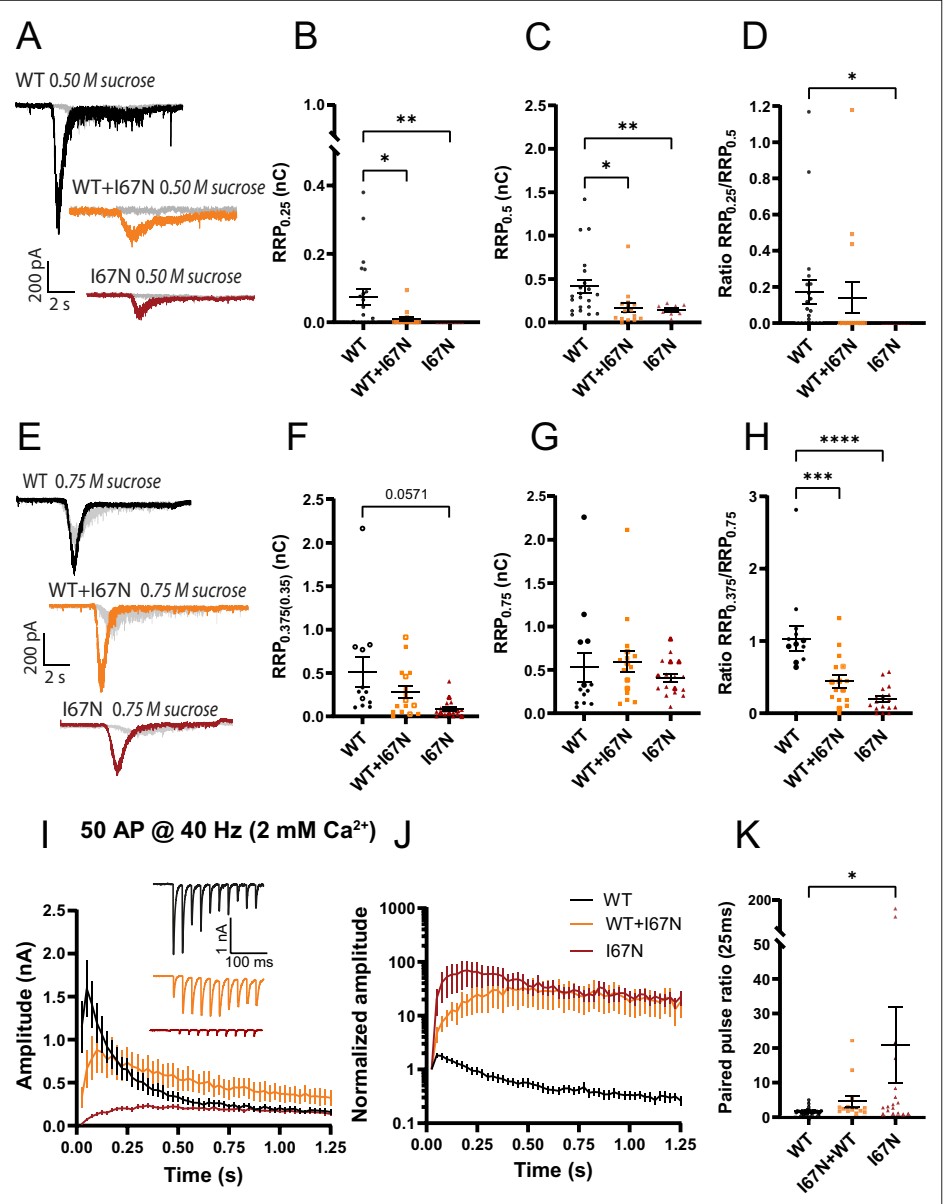

**Figure 11.** The I67N mutation has normal readily releasable pool (RRP) size, but increased energy barrier for fusion. (**A, E**) Example traces for the wildtype (WT), mutant, and co-expressed condition. Each cell was stimulated by 0.25 M (**A**, in gray) and 0.5 M sucrose (**A**, in color) or 0.375 M sucrose (**E**, in gray) and 0.75 M (**E**, in color). The charge released by 0.25 M sucrose (**B**, I67N: $n$ = 21, 15, 8 for WT, co-expressed, and mutant conditions, respectively) or 0.375 M sucrose (**F**, I67N: $n$ = 12, 16, 18; a few cells were stimulated with 0.35 M sucrose – shown with open symbols). (**B**) **$p < 0.01$; *$p < 0.05$, Kruskal–Wallis test with Dunn's multiple comparison test; (**F**) $p$ = 0.0339 Brown–Forsythe analysis of variance (ANOVA) test; Dunnett's multiple comparison test, $p = 0.0571$. The charge released by 0.5 M sucrose (**C**, I67N: $n$ = 21, 15, 8 for WT, co-expressed, and mutant conditions, respectively), or 0.75 M sucrose (**G**, I67N: $n$ = 13, 16, 18). (**C**) **$p < 0.01$, *$p < 0.05$, Brown–Forsythe ANOVA with Dunnett's multiple comparisons test. The ratio of the 0.25 and 0.5 M sucrose pool (**D**, I67N: $n$ = 21, 15, 8 for WT, co-expressed, and mutant conditions, respectively), or the ratio of 0.375 and 0.75 M sucrose pool (**H**, $n$ = 13, 16, 18). (**D**) *$p < 0.05$, Kruskal–Wallis test with Dunn's multiple comparisons test. (**H**) ****$p < 0.0001$; ***$p < 0.001$, ANOVA with Dunnett's multiple comparisons test. (**I**) eEPSCs in response to 50 APs at 40 Hz recorded in 2 mM extracellular $Ca^{2+}$ (I67N: $n$ = 23, 16, 20 for WT, co-expressed, and mutant conditions, respectively). Inserts: Normalized eEPSC amplitudes demonstrating facilitation of mutant conditions. (**J**) Normalized eEPSC amplitudes in response to 50 APs at 40 Hz recorded in 2 mM extracellular $Ca^{2+}$. (**K**) Paired-pulse ratio at interstimulus interval 25 ms (I67N: $n$ = 24, 14, 17 for WT, co-expressed, and mutant conditions, respectively). *$p < 0.05$, ANOVA with Dunnett's multiple comparison test. Source Data containing quantitative data are found in the Source Data files for *Figures 5 and 7*.

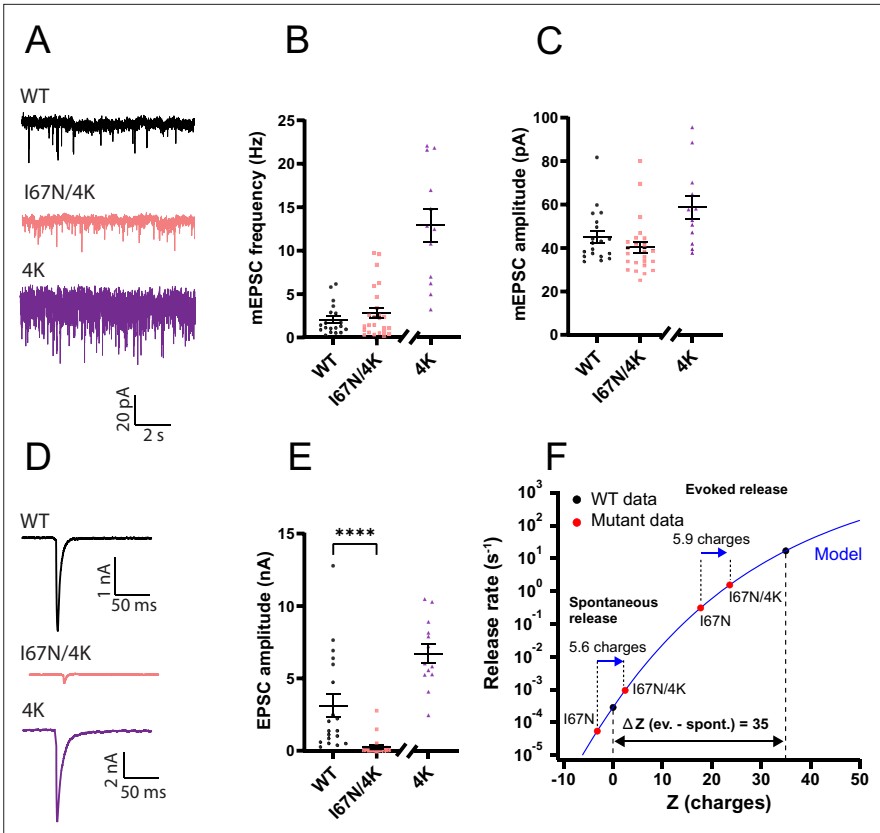

**Figure 12.** Adding positive surface charges to the SNARE complex partly compensate for the I67N mutation. (**A**) Example traces of mEPSC release for wildtype (WT), I67N/E183K/S187K/T190K/E194K (I67N/4K) and E183K/ S187K/T190K/E194K (4K) SNAP25b. Data from the 4K mutation were obtained in a separate experiment and are shown for comparison, but statistical tests with 4K mutation data were not carried out. (**B**) The mini frequencies for the WT and I67N/4K are not significantly different; data from the 4K mutation are shown for comparison ($n$ = 19, 25, 13 for WT, I67N/4K, and 4K, respectively). (**C**) Mini amplitudes remain unaffected by I67N/4K mutation. eEPSC examples (**D**) and amplitudes (**E**) for WT and I67N/4K; 4K is shown for comparison ($n$ = 19, 25, 13 for WT, I67N/4K, and 4K, respectively). ****$p$ < 0.0001 Mann–Whitney test. (**F**) Electrostatic triggering model (blue line; *Ruiter et al., 2019*) refitted to WT spontaneous and evoked data points (black points). Fitted parameters: rate 0.00029 s$^{-1}$ at zero (0) charge ($Z$); fraction $f$ = 0.030; the maximum rate was fixed at 6000 s$^{-1}$. WT (black points), I67N, I67N/4K (red points): means of log-transformed data. The charge values ($Z$, horizontal axis) for I67N and I67N/4K were found by interpolation in the model; the two spontaneous points (I67N, I67N/4K) are separated by 5.6 charges. For evoked release, rates were found by deconvolution and normalizing to RRP$_{0.5}$ (*Ruiter et al., 2019*). The $Z$-values for evoked release were found by interpolation in the model; the two mutants (I67N, I67N/4K) are separated by 5.9 charges.

The online version of this article includes the following source data for figure 12:

**Source data 1.** Excel file containing quantitative data.

were separated by 5.9 charges (***Figure 12F***). The fact that these numbers are close to the nominal 6 charges introduced shows that even for the I67N mutation, the same basal electrostatic model still applies, but the deleterious effect of the I67N on evoked release rates is larger than on spontaneous rates and therefore the rescue of evoked release rate by positive charges is insufficient to reach WT values. Moreover, because of the saturating form of the curve (i.e. the model) adding positive charges is an effective way of rescuing spontaneous, but not evoked release.

Overall, the I67N disease mutation increases the amplitude of the energy barrier for fusion, and it does so more for evoked than for spontaneous release, but the electrostatic mechanism, which we assume is part of release triggering (***Ruiter et al., 2019***), appears to be intact.

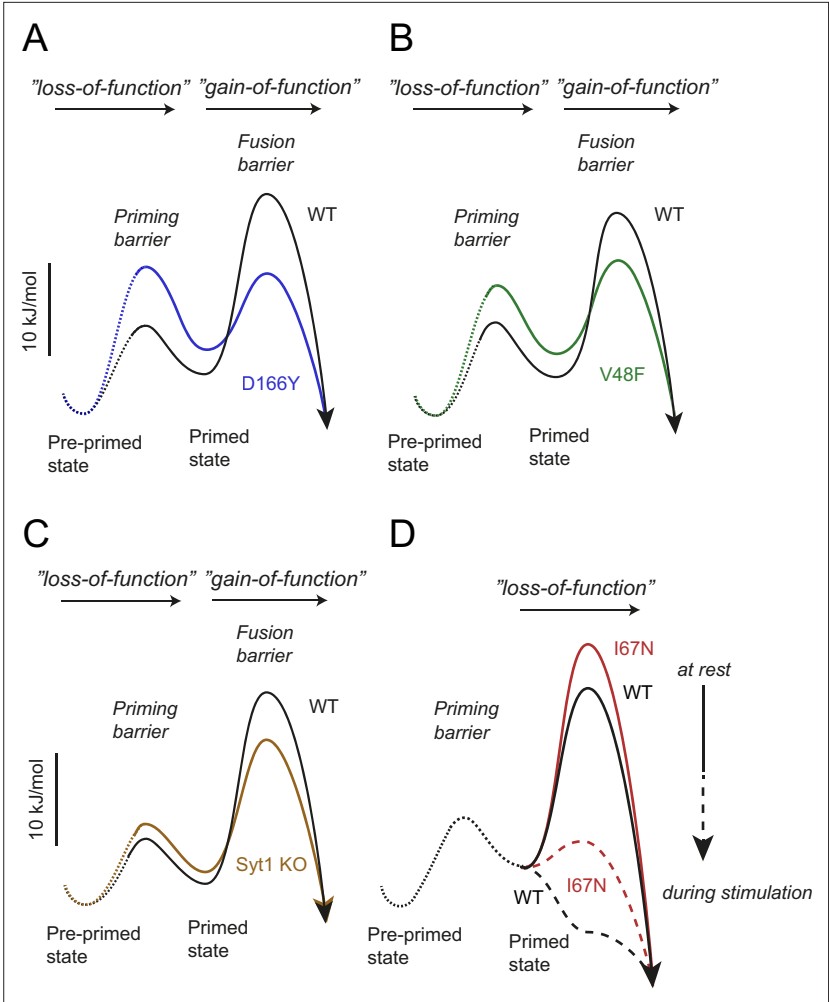

**Figure 13.** Energy landscapes. The energy landscapes of wildtype (WT) and mutants were calculated as explained in Materials and methods and displayed to scale. Energy landscapes for D166Y (**A**), V48F (**B**), and *Syt1* knockout (KO) (**C**) are shown at rest and are characterized by a higher priming barrier ('loss-of-function' phenotype), a destabilized readily releasable pool (RRP), and a lower fusion barrier ('gain-of-function' phenotype). The I67N (**D**) is characterized by a higher fusion barrier ('loss-of-function' phenotype). The relative increase in the fusion barrier by the I67N mutation is higher during stimulation than at rest. Dotted lines represent energy levels for which less is known.

## Discussion

We have shown that two SNAP25 mutations (V48F and D166Y) that compromise interaction with Syt1 lead to complex phenotypes characterized by a combination of loss-of-function and gain-of-function features. Thereby, the mutations fall into the 'neomorph' category, where the mutated protein has novel or changed interactions or functions (***Verhage and Sørensen, 2020***). In contrast, the I67N substitution within the SNARE bundle is a dominant-negative mutation.

### SNAP25 disease mutations change protein–protein interactions and the energy landscape of fusion

Both the V48F and the D166Y resulted in a decrease in the amplitude of the apparent energy barrier for fusion, whereas the I67N increased the amplitude of the apparent fusion barrier. A vesicle's release willingness can only be assessed by fusing it; therefore, it is not possible to distinguish between effects on the fusion barrier per se, and effects on the fusion machinery. In recognition of this fact, we here refer to the 'apparent energy barrier'. Using Arrhenius' equation to convert fusion, priming and depriming rates to their respective energy barrier heights (see Materials and methods for the

assumptions behind this procedure), we can derive the energy landscape for fusion of the three mutants (*Figure 13A–D*). This shows the multiple changes in the V48F and D166Y, which affect at least two different barriers (priming and fusion, *Figure 13A, B*), leading to a complex phenotype, whereas for I67N the fusion barrier is primarily (or solely) affected (*Figure 13D*).

When combining the I67N with the 4K mutation, which introduces six extra positive fixed charges, we could place our data within the framework of our model for electrostatic triggering (*Ruiter et al., 2019*) and show that the effect of charge per se is approximately the same in the I67N mutant as in the WT. Note that there are endogenous positively charged amino acids toward the C-terminal end of SNAP25 that are important for release rates (*Fang et al., 2015*). Rescue of spontaneous release was completed by adding six positive charges, which is consistent with the idea that the assembly of the C-terminal end of the SNARE complex, which is compromised by I67N (*Rebane et al., 2018*), works against the electrostatic energy barrier, which is affected by the SNARE surface charge (*Ruiter et al., 2019*). In contrast, rescue of evoked release by charges were incomplete, due to the larger effect of I67N on evoked release, combined with the shallow effect of charges on evoked release (*Ruiter et al., 2019*). The larger susceptibility of evoked release to C-terminal mutation of the SNAREs might be partly due to the higher number of SNARE complexes involved in evoked than in sustained/spontaneous release (*Mohrmann et al., 2010*).

The effect of the I67N mutation in the energy domain at rest can be calculated from the spontaneous release rate, which was reduced by a factor 22.4 (from $1.31 \pm 0.36$ to $0.0583 \pm 0.0274$ Hz). This corresponds to an effect in the energy domain of 3.1 $k_BT$ (where $k_B$ is Boltzmann's constant; assuming unchanged RRP size, pool normalization is not required). Work with single-molecular optical tweezers showed that the I67N mutation destabilizes the overall SNARE C-terminal and linker domain, which are supposed to deliver the power stroke for membrane fusion, by 14 $k_BT$ (*Rebane et al., 2018*), which is substantially more. The reduction in spontaneous release rate is more comparable to the reduction in the transition rate of folding by the C-terminal and linker domain by a factor of ~10 (*Rebane et al., 2018*). Since at least three SNARE complexes, possibly more, contribute to vesicle fusion (*Bao et al., 2018*; *Manca et al., 2019*; *Mohrmann et al., 2010*; *Shi et al., 2012*), folding kinetics correlates better to spontaneous fusion rates than overall SNARE complex stability.

The D166Y and V48F mutations lead to increases in spontaneous release, and more asynchronous eEPSCs, consistent with their localization in the primary SNARE:Syt1 interface (*Schupp et al., 2016*; *Zhou et al., 2015*; *Zhou et al., 2017*), and the demonstrated impaired Syt1 binding (*Figure 8B, C*), see also *Alten et al., 2021*. These phenotypes are at first glance similar to Syt1 knockout/knockdown (*Bouazza-Arostegui et al., 2022*; *Chang et al., 2018*; *Huson et al., 2020*; *Ruiter et al., 2019*). However, when normalized to the RRP size, mEPSC frequencies were much higher in the V48F and the D166Y than in the *Syt1* KO (*Figure 6D*), and dual sucrose applications indicated a decrease in the amplitude of the sucrose-probed apparent energy barrier and increased release probability, features not found in the *Syt1* KO (*Bouazza-Arostegui et al., 2022*; *Huson et al., 2020*). This indicates a gain-of-function feature of these mutations, which fulfills D166Y > V48F; such a feature was identified as an increased interaction of V48F and D166Y with VAMP2- and syntaxin-1-containing SUVs (*Figure 9D*); the interaction was stronger for D166Y than for V48F. Molecular dynamics (MD) simulations of the SNAP25 mutants did not reveal major structural changes in the SNAP25 backbone compared to WT (*Figure 9—figure supplement 2A–C*). Nevertheless, the calculation of electrostatic (Coulomb) and van der Waals (Lennard–Jonson, LJ) interactions for residues 48–52 (*Figure 9—figure supplement 2D*) and residues 162–166 (*Figure 9—figure supplement 2E*) shows that for D166Y the interaction energy is substantially more negative. This suggests that the interaction between D166Y and nearby residue H162 is stronger than in WT. Overall, this may result in stabilization of a structure consistent with SNARE complex formation and explain why D166Y is a stronger gain-of-function mutation than V48F. However, note that the AlphaFold prediction (*Jumper et al., 2021*; *Mirdita et al., 2022*) used as a starting point for these simulations is identical to the structure of SNAP25 in the assembled SNARE complex (*Zhou et al., 2015*), whereas unassembled SNAP25 is likely less structured or unstructured (*Fasshauer et al., 1997*). Overall, these disease mutants do not only fail in their interaction with Syt1, they bypass fusion control, resulting in premature SNARE complex assembly. Although the increased spontaneous release rate might cause patient symptoms, *Alten et al., 2021* suggested that the resulting postsynaptic depolarization might compensate for the smaller eEPSC amplitude to normalize the overall firing rate.

In an SUV:GUV fusion assay, where folding of SNAP25 onto syntaxin-1/VAMP2 is rate limiting, D166Y and V48F caused an increase in pre-stimulation fusion rates. This is consistent with recent data showing that folding of SNAP25 onto a template formed by VAMP2 and syntaxin-1 held in place by Munc18-1 might be a late, rate-limiting, step in exocytosis (*Jiao et al., 2018*). This process is regulated and sped up by Munc13-1 (*Kalyana Sundaram et al., 2021*; *Shu et al., 2020*; *Wang et al., 2019*). The fact that a similar effect on spontaneous fusion was seen in the assay with preformed t-SNAREs in the presence of complexin indicates that there is an assembly step, even in that assay, which can be sped up by the mutations. This aligns with the demonstration by single-molecule FRET of a further assembled (tighter) state of the trans-SNARE complex induced by $Ca^{2+}$-unbound Syt1, which becomes committed for fusion once $Ca^{2+}$ binds (*Das et al., 2020*).

## V48F and D166Y change the size of the RRP via effects on priming, depriming and fusion

We found that V48F and D166Y cause a decrease in RRP size, whether measured by sucrose application or by train stimulation. The reduced RRP appears to be at variance with the publication by *Alten et al., 2021*, who used sucrose application to larger mixed cultures. There might be several reasons for this discrepancy. First, Alten et al. used longer duration of sucrose application (~1 min) and applied sucrose to a large mixed culture, which might result in a variable delay such that all synapses are not stimulated simultaneously. This will result in temporally overlapping release of RRP and upstream vesicle pools, which then cannot be distinguished from each other. When grown on 50 µm micro-islands, sucrose can be applied acutely (within ~0.05 s) to the entire dendritic tree and all synapses using a local perfusion system, which allows distinguishing the RRP from upstream pools. Second, Alten assessed the RRP for GABAergic synaptic transmission in mixed culture, whereas we measured the RRP for glutamatergic neurons in autaptic culture. The RRP has a different substructure in the two types of neurons (*Moulder and Mennerick, 2005*), which might lead to different findings. Third, it was recently shown that sufficient neuronal maturation is necessary to detect the decrease in RRP upon Syt1 elimination (*Bouazza-Arostegui et al., 2022*), which is likely modulated by the presence of other neurons (*Chang et al., 2018*; *Liu et al., 2009*; *Wierda and Sørensen, 2014*). This adds an additional layer of complexity since neuronal maturation likely varies between laboratories.

Dissection of the three rates that determine RRP size (priming, depriming, and fusion rates) showed that all three are changed in the V48F and D166Y mutations. Especially for D166Y, but also for V48F, spontaneous release contributed to RRP depletion by triggering premature fusion. Significant RRP depletion has not been expected for moderate increases in mEPSC frequency (*Rhee et al., 2005*; *Ruiter et al., 2019*), based on the argument that RRP refilling should be fast enough to counteract depletion. However, to properly make this argument, priming, depriming, and spontaneous release rates must be compared in the steady-state situation. Similarly, we recently showed by cryo-electron tomography that the 4K mutation, which increased the mEPSC frequency to ~30 Hz and had a reduced RRP (*Ruiter et al., 2019*) caused a loss of synaptic vesicles tethered to the membrane with three tethers (*Radecke et al., 2023*), which is the structural correlate of the RRP (*Fernández-Busnadiego et al., 2010*).

The major effect on RRP size is caused by a reduction in forward priming rate by D166Y and V48F. Comparison to the Syt1 KO showed qualitatively similar changes, but of a smaller magnitude, with spontaneous release playing a negligible role for RRP size. Notably, in all three cases a reduction in depriming rate partly counteracted the lowered priming rate (*Table 1*) – this was only significant for D166Y. Indeed, part of the role of Syt1:SNARE interaction might be catalytic, lowering the energy level of a transition state along the path to priming, which will affect both rates (*Walter et al., 2013*). This might happen because transient binding to Syt1 might structure SNAP25 and assist in formation of the SNARE complex, whereas SNAP25 mutants might prestructure the protein, bypassing the need for Syt1. In an energy diagram (*Figure 13*), it becomes clear that vesicle priming and regulation of spontaneous release are interdependent. Stabilization of the RRP state will both increase RRP size at rest and reduce spontaneous release, since lowering the energy level of (i.e. stabilizing) RRP vesicles will increase the size of the energy barrier that the RRP vesicles face (*Figure 13*). Consistently, down-regulation (clamping) of spontaneous release and upregulation of evoked release are often interdependent under conditions where Syt1 expression level, or Syt1 interaction with the SNAREs, are up- or downregulated (*Courtney et al., 2021*; *Courtney et al., 2019*; *Vevea and Chapman, 2020*; *Zhou*

*et al., 2015*; *Zhou et al., 2017*). This does not rule out the existence of mutations that can affect one mode of release more than the other (e.g. the I67N/4K mutation). By inference, assembly of the primary Syt1:SNARE interface (*Zhou et al., 2017*) is most likely involved in both clamping release and setting up an RRP. In further support of this, the minimal in vitro assay with preassembled t-SNARE dimers displayed a qualitatively similar reduction in calcium-dependent release, with D166Y being more impaired than V48F (*Figure 8D*). Overall, the energetic contribution of Syt1:SNARE interaction to Ca$^{2+}$-triggered release is a stabilization of upstream steps, and the increase of the fusion barrier downstream of the RRP (*Figure 13*). The electrostatic nature of the fusion barrier (*Ruiter et al., 2019*) ensures its rapid dissolution by Ca$^{2+}$, possibly by unbinding of Syt1 from the SNARE complex (*Voleti et al., 2020*).

For the I67N mutation, the sucrose RRP was also reduced in size when using 0.5 M sucrose, which is consistent with previous observations (*Alten et al., 2021*). However, when the energy barrier for release is increased, 0.5 M sucrose is insufficient to deplete the RRP (*Schotten et al., 2015*). Accordingly, 0.75 M sucrose released an RRP of similar size in WT, I67N, and I67N + WT co-expressing cells, showing that the vesicle priming reaction is intact, but the vesicles face a larger fusion barrier (*Figure 11*). In this mutant, train stimulations in the I67N or co-expressed WT + I67N case resulted in a phenotype quite distinct from the WT, with strong facilitation throughout the train. Thus, the I67N is a strongly dominant-negative mutant, whether considering mEPSC frequency, eEPSC amplitude, or train stimulations, whereas V48F and D166Y are incompletely dominant. This can be explained by the different function of SNAP25 domains during fusion, where V48 and D166 help set up an arrested primed vesicle state by interacting with Syt1 (*Schupp et al., 2016*; *Zhou et al., 2015*; *Zhou et al., 2017*), whereas I67 participates in the final conformational change, the 'stroke' that leads to assembly of the C-terminal end of the complex and the linker domain. Due to their defect in priming, V48F and D166Y might not enter the super-complex (the complex of SNARE complexes driving fusion) as often as WT protein. In contrast, I67N will readily enter the super-complex and compromise its function, which will lead to a dominant-negative phenotype due to the multiple SNARE complexes involved in fusion (*Bao et al., 2018*; *Mohrmann et al., 2010*; *Shi et al., 2012*).

Back-extrapolation of 40 Hz trains (@4 mM Ca$^{2+}$) also led to the conclusion that V48F and D166Y have a reduced RRP$_{ev}$, the RRP sub-pool that action potentials draw on, and a reduced forward priming rate. Interestingly, the release probability when normalized to the sucrose pool (RRP$_{0.5}$) was increased non-significantly for the V48F and significantly for the D166Y mutation, but when considering 40 Hz trains, the release probabilities of both mutations were decreased, consistent with a shift toward facilitation. The latter finding is likely caused by the defective interaction with Syt1, which leads to suboptimal priming and/or defective super-priming, which is an additional priming step after entry of the vesicle into the RRP (*Lee et al., 2013*; *Taschenberger et al., 2016*). This will lead to reduction of the first eEPSC of a train and thereby a lowered nominal release probability. However, when comparing the eEPSC charge to the RRP$_{0.5}$, the strong reduction in the RRP$_{0.5}$ pool (especially in the D166Y mutation) accounts for the increase in the overall release probability. This can be explained if spontaneous release causes a disproportional depletion of vesicles, which are in the RRP$_{0.5}$, but not in the RRP$_{ev}$.

## Conclusion

SNAP25 disease missense mutations change the function of the protein without compromising its expression, leading to dominant negative or neomorphic mutations. Missense mutations in the primary SNARE:Syt1 interface (V48F and D166Y) result in a complex phenotype characterized by loss-of-function in the priming step and gain-of-function in the fusion step. Missense mutation in the SNARE bundle (I67N) leads to an increased amplitude of the energy barrier for fusion. In addition, disease mutations display inefficient rescue of neuronal survival. Overall, SNAP25 encephalopathy caused by single missense point mutations presents with interdependent functional deficits, which must be overcome for successful treatment.

## Materials and methods
### Animals

*Snap25* KO (*Washbourne et al., 2002*) and *Synaptotagmin-1* (*Syt1*) KO (*Geppert et al., 1994*) C57BL/6 mice: Heterozygous animals were routinely backcrossed to C57BL/6J to generate new

heterozygotes. The strain was kept in the heterozygous condition and timed pregnancies were used to recover KO embryos by cesarean section at embryonic day 18 (E18). Pregnant females were killed by cervical dislocation; E18 embryos of either sex were collected and killed by decapitation. Permission to keep, breed, and use *Snap25* and *Syt1* mice was obtained from the Danish Animal Experiments Inspectorate (2018-15-0202-00157) and followed institutional guidelines as overseen by the Institutional Animal Care and Use Committee (IACUC). CD1 outbred mice were used to create astrocytic cultures and mass cultures for Western blotting. Newborns (P0–P2) of either sex were used. Pups were killed by decapitation.

## Cell lines

HEK293-FT cells for production of lentiviruses were obtained from the Max-Planck-Institute for biophysical chemistry. The cells were passaged once a week, and they were used between passages 11 and 25 for generation of lentiviral particles. The cells were kept in Dulbecco's modified Eagle medium (DMEM) + Glutamax (Gibco, cat. 31966047) supplemented with fetal bovine serum (Gibco, cat. 10500064), Pen/Strep (Gibco, cat. 15140122), and Geneticin G418 (Gibco, cat. 11811064) at 37°C in 5% $CO_2$. Negative mycoplasma status was confirmed using a commercial kit (Venor GeM OneStep, Minerva biolabs, Art. Nr. 11-8025).

## Preparation of neuronal culture

Self-innervating (autaptic) hippocampal cultures were used (*Bekkers and Stevens, 1991*). Astrocytes were isolated from CD1 outbred mice (P0–P2) of either sex. The cortices were isolated from the brains in 4-(2-hydroxyethyl)-1-piperazineethanesulfonic acid (HEPES)-buffered Hank's Balanced Salt Solution (HBSS–HEPES) medium (HBSS supplemented with 1 M HEPES) and the meninges were removed. The cortical tissue was chopped into smaller fragments, transferred to 0.25% trypsin dissolved in DMEM solution (450 ml Dulbecco's MEM with 10% fetal calf serum, 20,000 IU penicillin, 20 mg streptomycin, 1% MEM non-essential amino acids), and incubated for 15 min at 37°C. Subsequently, inactivation medium (12.5 mg albumin +12.5 mg trypsin-inhibitor in 5 ml 10% DMEM) was added, the tissue was washed with HBSS–HEPES, triturated and the cells were plated in 75 cm² flasks with prewarmed DMEM solution (one hemisphere per flask) and stored at 37°C with 5% $CO_2$. Glial cells were used after 10 days.

Glass coverslips were washed overnight in 1 M HCl; for an hour in 1 M NaOH and washed with water before storage in 96% ethanol. Coverslips were first coated by 0.15% agarose and islands were made by stamping the coating mixture (3 parts acetic acid [17 mM], 1 part collagen [4 mg/ml], and 1 part poly-D-lysine [0.5 mg/ml]) onto the glass coverslips using a custom rubber stamp. Glial cells were washed with prewarmed HBSS–HEPES. Trypsin was added and the flasks were incubated at 37°C for 10 min. Cells were triturated and counted in a Bürker chamber before plating onto the glass coverslip with DMEM solution. After 2–5 days, neurons were plated on the islands.

Neurons for autaptic culture were isolated from E18 *Snap25* KO mice of either sex. Pups were selected based on the absence of motion after tactile stimulation and bloated neck (*Washbourne et al., 2002*); the genotype was confirmed by Polymerase Chain Reaction (PCR) in all cases. The cortices were isolated from the brains in the HBSS–HEPES medium. The meninges were removed and the hippocampi were cut from the cortices before being transferred to 0.25% trypsin dissolved in HBSS–HEPES solution. The hippocampi were incubated for 20 min at 37°C, subsequently washed with HBSS–HEPES, triturated and the cell count was determined with a Bürker chamber before plating on the islands (7000–8000 neurons per well). Cells were incubated in the NB medium (Neurobasal with 2% B-27, 1 M HEPES, 0.26% Glutamax, 14.3 mM β-mercaptoethanol, 20,000 IU penicillin, 20 mg streptomycin) and used for measurements between DIV10 and 14.

Hippocampal neurons for high-density cell culture for Western blotting were obtained from P0 to P1 CD1 outbred mice. The dissection, tissue digestion, and cell counting were performed the same way as the neurons for the autaptic culture, the high-density culture (600,000 neurons per well) was then kept in NB-A medium (Neurobasal-A with 2% B-27, 1 M HEPES, 0.26% Glutamax, 20,000 IU penicillin, 20 mg streptomycin) and half the volume of the media was replaced every 2–3 days before harvesting the cells at DIV14.

## Constructs for rescue experiments

SNAP25b was N-terminally fused to GFP and cloned into a pLenti construct with a CMV promoter (*Delgado-Martínez et al., 2007*). Mutations were made using the QuikChange II XL kit (Agilent).

Primers were ordered from TAGC Copenhagen. All mutations were verified by sequencing before virus production. Viruses were prepared as previously described using transfection of HEK293FT cells (*Naldini et al., 1996*). Neurons were infected with lentiviruses on DIV 0–1, 30 µl total virus per well (WT or mutant; 15 µl + 15 µl WT + mutant virus for co-expressed condition).

## Immunostaining and confocal microscopy

Autaptic hippocampal neurons were fixed at DIV14 in 2% paraformaldehyde (PFA) in culture medium for 10 min and subsequently in 4% PFA for 10 min. Cells were then washed with phosphate-buffered saline (PBS), permeabilized by 0.5% Triton X-100 for 5 min, and blocked with 4% normal goat serum in 0.1% Triton X-100 (blocking solution) for 30 min. Cells were incubated with primary antibodies diluted in blocking solution (anti-MAP2, 1:500, chicken, ab5392, Abcam; and anti-vGlut1 1:1000, guinea pig, AB5905, Merck Millipore) for 2 hr at room temperature (RT). After washing with PBS, the cells incubated with secondary antibodies in blocking solution for 1 hr at RT in the dark (anti-chicken Alexa 568, 1:1000, A11041, Thermo Fisher Scientific; and anti-guinea pig Alexa 647, 1:1000, A-21450, Thermo Fisher Scientific) and washed again. Coverslips were mounted with FluorSave and imaged on Zeiss CellObserver spinning disc confocal microscope (×40 water immersion objective; NA 1.2) with Zeiss Zen Blue 2012 software. Images were acquired as Z-stack and 9 images per plane to capture the whole island in the field of view. The images were post-processed with Zeiss Zen Black software and neuronal morphology was analyzed using SynD automated image analysis (*Schmitz et al., 2011*).

## Western blotting

Harvesting the high-density hippocampal culture, Bicinchoninic acid (BCA) assay and transferring the protein samples on a polyvinylidene difluoride (PVDF) membrane were performed as described before (*Ruiter et al., 2019*). Incubation in primary antibodies (a-SNAP25: mouse, 1:10,000, SYSY 111011, Synaptic Systems; a-VCP: mouse, 1:2000, ab11433, Abcam) was performed overnight with 70 rpm shaking at 4°C, followed by washing in Tris-buffered saline with 0.1% Tween 20 Detergent (TBST) and a 1-hr incubation in secondary antibody (goat a-mouse-HRP: 1:10,000, P0447, Dako). After washing, Pierce ECL Western blotting substrate was added and chemiluminescence was visualized with FluorChem E (Proteinsimple). The Western blots were quantified using ImageJ (1.52a) to measure the signal intensity of the protein bands relative to the loading control (VCP). All relative levels of the target protein were normalized to the average relative level of the same target protein in the WT samples.

## Electrophysiology

Autaptic cultures were used from DIV10 until DIV14. The intracellular pipette medium contained: KCl 136 mM, HEPES 17.8 mM, creatine phosphate 15 mM, Na-ATP 4 mM, creatine phosphokinase 50 U, $MgCl_2$ 4.6 mM, Ethylene glycol tetraacetic acid (EGTA) 1 mM (pH 7.4, osmolarity ~300 mOsm). The standard extracellular recording medium contained: NaCl 140 mM, KCl 2.4 mM, HEPES 10 mM, glucose 14 mM, $CaCl_2$ 2 mM, $MgCl_2$ 2 mM. The extracellular recording medium for depriming experiments contained: NaCl 140 mM, KCl 2.4 mM, HEPES 10 mM, glucose 14 mM, $CaCl_2$ 4 mM, $MgCl_2$ 4 mM (pH 7.4, osmolarity ~300 mOsm). An Axio Observer A1 inverted microscope (Zeiss) was used to visualize the cells. The recordings were performed at room temperature. An EPC10 amplifier (HEKA) was used with the program Patchmaster v2.73 (HEKA). Traces were filtered with a 3-kHz Bessel low-pass filter and data were acquired at 20 kHz. The series resistance was compensated to 70%. Glass pipettes were freshly pulled on a P1000 pipette puller (Sutter Instruments) from borosilicate glass capillaries. Pipets ranging from 2.5 to 5 MΩ were selected for recordings. Cells with starting access resistance above 10 MΩ were rejected. Recordings were performed in voltage clamp, with the holding potential kept at −70 mV. Evoked excitatory postsynaptic currents (eEPSC) were induced by raising the holding voltage to 0 mV for 2 ms. Sucrose was dissolved into the standard extracellular recording medium. Application of the extracellular media was done using a custom-made barrel system, controlled by SF-77B perfusion fast step (Warner Instruments) controlled via digital output switches from the EPC10. Electrophysiological data were analyzed in IGOR Pro (v6.21 and v6.37, WaveMetrics) using a custom-written script. mEPSCs were analyzed with MiniAnalysis (v6.0.7, Synaptosoft).

## Electrophysiological data: calculations

Deconvolution was calculated and the electrostatic model for triggering was fitted to 4K and I67N/4K data as described before (*Ruiter et al., 2019*). Since the evoked release for I67N was so small that deconvolution became unreliable, we downscaled the peak release rate of the I67N/4K mutation with a factor of 5.616, corresponding to the reduction in eEPSC amplitude in the I67N compared to I67N/4K.

In order to determine the reasons for the reduced RRP size for V48F and D166Y, we considered a single pool model for the RRP, with priming rate $k_1$, depriming rate $k_{-1}$, and fusion rate $k_f$ (*Figure 6A*). The equation describing the evolution in RRP size is:

$$\frac{d\text{RPP}(t)}{dt} = k_1 - (k_{-1} + k_f) \cdot \text{RRP}(t) \tag{1}$$

where $k_1$ has the unit vesicles/s, whereas $k_{-1}$, and $k_f$ have the unit 1/s. The solution of this differential equation is

$$\text{RPP}(t) = \frac{k_1}{(k_{-1} + k_f)} \left(1 - e^{-(k_{-1}+k_f)t}\right) \tag{2}$$

The steady-state RPP size is

$$\text{RRP} = \frac{k_1}{(k_{-1} + k_f)} \tag{3}$$

The miniature release rate at steady state is

$$r_{mini} = \frac{k_f k_1}{(k_{-1} + k_f)} \tag{4}$$

When stimulated by sucrose, $k_f$ increases, and if $k_f \gg k_{-1}$ (for justification, see below) the current plateau will report on the forward priming rate alone:

$$r_{prime} \approx k_1 \tag{5a}$$

Thus, the current plateau during sucrose application can be used for estimating the forward priming rate. However, application of 0.5 M sucrose causes cell shrinkage and changes in solution viscosity, which in turn can change the leak current. This might cause the plateau current to change, which might be invisible during the experiment due to the synaptic events. To correct for this, we implemented variance-mean analysis to identify the true baseline current (the current corresponding to the lack of synaptic release). Synaptic release is essentially a source of shot noise, for which the variance is proportional to the mean. We therefore calculated the variance (after subtraction of a running average) and mean of the current in 50-ms intervals during the sucrose application, and performed linear regression in a variance-mean plot. The corrected baseline was identified by back-extrapolation to the variance level found in the absence of synaptic activity (using a stretch of current before sucrose application), as illustrated in *Figure 6C*.

If sucrose does not sufficiently increase $k_f$, *Equation 5a* would be replaced by

$$r_{mini} = \frac{k_f k_1 N_{suc}}{(k_{-1} + k_f N_{suc})} \tag{5b}$$

where $N_{suc}$ is the fold-increase in $k_f$ induced by sucrose application. Plotting the solutions to *Equations 3 and 4*, *Equation 5b* at different values of $N_{suc}$, the dependency of the estimated $k_{-1}$ and $k_1$ upon $N_{suc}$ can be investigated (*Figure 6—figure supplement 1*). Notably, in WT cells $N_{suc} > 5000$ (*Schotten et al., 2015*), indicating that the estimated values of $k_{-1}$ and $k_1$ using *Equation 5a* (*Table 1*) is quite accurate for the WT case, and the estimation in the case of the D166Y and V48F is even less dependent on $N_{suc}$, because the value of $k_f$ is higher at rest (*Figure 6—figure supplement 1*). Importantly, for no realistic value of $N_{suc}$ would the conclusion of decreased $k_1$ and $k_{-1}$ in the two mutants be in jeopardy.

For calculating the energy profiles of WT and mutants, we used Arrhenius' equation:

$$k = Ae^{\left(\frac{-E_A}{RT}\right)} \tag{6}$$

where $E_A$ is the activation energy, $R$ and $T$ are the gas constants and the absolute temperature, respectively, and $A$ is an empirical constant that depends on collision rates (*Schotten et al., 2015*). Solving for the activation energy, we get:

$$E_a = RT \left(\ln\left(A\right) - \ln\left(k\right)\right) \tag{7}$$

Since $A$ is unknown, we cannot use this equation to calculate the absolute values of the transition energies; however, when we compare a mutation to the WT condition, and under the assumption that A is unchanged by mutation, we can calculate the difference in energy level of the transition states:

$$\Delta E_a^{Mut-WT} = RT \left(\ln\left(k^{WT}\right) - \ln\left(k^{Mut}\right)\right) \tag{8}$$

Using this equation sequentially, for the priming rate, the depriming rate and the fusion rate, we can derive the entire energy diagram, under the additional assumption that the energy in the pre-primed state is identical between WT and mutation, and using that at room temperature, $RT = 2.479$ kJ/mol.

The assumption that the empiric factor, $A$, is unchanged by mutation is likely to hold for the fusion reaction, which depends on conformational changes in a preformed complex. In contrast, collision rates might be involved in priming; in that case, the effect of the V48F and D166Y mutations on priming, which we here attribute to an increase in the energy level of the priming transition state might reflect a lower collision rate between vesicles and plasma membrane fusion machinery, and/or a lower energy level of the pre-primed state.

## Constructs for in vitro protein expression

The following constructs were used: Glutathione S-Transferase (GST) – full-length VAMP2 is encoded by plasmid pSK28 (*Kedar et al., 2015*), GST-cytosolic domain VAMP2 (amino acids 1–94) (pSK74, *Ruiter et al., 2019*), synaptotagmin 1-His6 lacking the luminal domain (amino acids 57–421) (pLM6, *Mahal et al., 2002*), His6-complexin II (CpxII) (pMDL80, *Malsam et al., 2012*), His6-syntaxin-1A (pSK270), His6-SNAP25b (pFP247, *Parlati et al., 1999*), t-SNARE consisting of syntaxin-1A and His6-SNAP25b (pTW34, *Parlati et al., 1999*). To generate the His6-syntaxin-1A (pSK270) construct, the full-length syntaxin-1A sequence with an N-terminal His6-tag was subcloned into a pETDuet1 vector. Tag-removal with prescission protease results in an N-terminal extension of three additional amino acids (Gly–Pro–Gly) preceding the N-terminus of the syntaxin-1 sequence. Point mutants in soluble His6-SNAP25b and in the t-SNARE complex were generated by using the DNA templates pFP247 and pTW34, respectively, and the Quikchange DNA mutagenesis kit (QIAGEN) (*Ruiter et al., 2019*). Thereby, the following SNAP25 constructs were established: His6-SNAP25b mutant I67N (pUG1), V48F (pUG2), D166Y (pUG3), and t-SNAREs containing the corresponding SNAP25b mutants I67N (pUG7), V48F (pUG8), and D166Y (pUG9). The identity of all constructs was validated by DNA sequencing.

## Protein expression and purification

In general, the expression vectors, encoding the desired protein constructs were transfected into *Escheria coli* BL21 (DE3) (Stratagene). At an OD$_{660}$ of 0.8, protein expression was induced by the addition of 0.3 mM Isopropyl β-D-1-thiogalactopyranoside (IPTG). Alternatively, proteins were expressed by autoinduction using buffered media containing lactose (*Studier, 2005*). Cells were harvested by centrifugation (3500 rpm, 15 min, H-12000 rotor, Sorvall) and lysed using the high-pressure pneumatic processor 110 L (Microfluidics). Cell fragments were removed by centrifugation at 60,000 rpm (70Ti rotor, Beckman) for 1 hr and the clarified supernatant was snap frozen in liquid nitrogen.

The purification of full-length VAMP2 was performed as described previously (*Kedar et al., 2015*) with the following modifications. Cells were grown in ZYM media (*Studier, 2005*) and protein expression was induced with IPTG for 3 hr at 25°C. The purification and expression of the GST-tagged cytosolic domain of VAMP2 were described previously (*Ruiter et al., 2019*). Synaptotagmin-1-His6 lacking the luminal domain was purified as described previously (*Malsam et al., 2012*) with the

following modification. After dilution to 50 mM salt, the protein was further purified on a MonoS Sepharose column (GE Healthcare) applying a gradient of 50–500 mM KCl in 25 mM HEPES–KOH (pH 7.4).

His6-syntaxin-1A purification was performed as outlined by *Schollmeier et al., 2011* with the following modifications. Briefly, cells were grown in ZYM media (*Studier, 2005*) and autoinduction was used for the protein expression at 22°C overnight. Syntaxin-1A was eluted from Ni-NTA beads (QIAGEN) by overnight cleavage with Prescission protease (GE Healthcare) at 4°C, removing the His6 tag. After dilution to 80 mM salt, the protein was further purified on a MonoQ Sepharose column (GE Healthcare) applying a gradient of 50–500 mM KCl in 25 mM HEPES–KOH (pH 7.4).

His6-SNAP25 was expressed as depicted for syntaxin-1A and purified via Ni-NTA beads, followed by MonoQ Sepharose column chromatography (*Ruiter et al., 2019*). Preassembled full-length t-SNARE complexes were expressed and purified as described previously (*Weber et al., 1998*). His6-Complexin II expression and purification were performed according to *Malsam et al., 2012* with the following modifications. His6-CpxII was expressed in BL21-DE3 codon + bacteria for 2 hr at 27°C.

The concentrations of purified proteins were determined by sodium dodecyl sulfate–polyacrylamide gel electrophoresis (SDS–PAGE) and Coomassie Blue staining using Bovine Serum Albumin (BSA) as a standard and the Fiji software for quantification. Furthermore, mutant and WT protein concentrations were directly compared on a single gel.

## Protein reconstitution into liposomes

All lipids were from Avanti Polar Lipids, except of Atto488-DPPE and Atto550-DPPE, which were purchased from ATTO-TEC. For VAMP2 and Syt1 reconstitution into SUVs, lipid mixes (3 µmol total lipid) with the following composition were prepared: 25 mol% POPE (1-hexadecanoyl-2-octadecenoyl-*SN*-glycero-3-phosphoethanolamine), 15 mol% DOPS (1,2-dioleoyl-*SN*-glycero-3-phosphoserine), 29 mol% POPC (1-palmitoyl-2-oleoyl-*SN*-glycero-3- phosphocholine), 25 mol% cholesterol (from ovine wool), 5 mol% PI (L-α-phosphatidylinositol), 0.5 mol% Atto488-DPPE (1,2-dipalmitoyl-SN-glycero-3-phosphoethanolamine), and 0.5 mol% Atto550-DPPE. For docking assays, the t-SNARE liposome lipid mix (5 µmol total lipid) had the following composition: 35 mol% POPC, 15 mol% DOPS, 20 mol% POPE, 25 mol% cholesterol, 4 mol% PI, 0.05 mol% Atto647-DPPE, and 0.5 mol% tocopherol. For the preparation of PI(4,5)P2-containing t-SNARE and syntaxin-1A liposomes, the t-SNARE liposome lipid mix was used, but 1 mol% PI(4,5)P2 (L-α-phosphatidylinositol-4,5-bisphosphate) was added and the POPC concentration lowered by 1% accordingly.

Proteins were reconstituted as described previously (*Malsam et al., 2012*). For the docking and fusion assays, t-SNARE WT and mutants were reconstituted at a protein to lipid ratio of 1:900. For the syntaxin-1A membrane fusion assay, syntaxin-1A was reconstituted at a protein to lipid ratio of 1:1000. Briefly, 5 µmol dried lipids were dissolved in 0.7 ml reconstitution buffer (25 mM HEPES–KOH, pH 7.4, 550 mM KCl, 1 mM Ethylenediaminetetraacettic acid (EDTA–NaOH), 1 mM dithiothreitol (DTT)) containing final 1.4 wt% octyl-β-D-glucopyranoside (β-OG) and either 6.5 nmol (390 µg) t-SNARE complex or 5 nmol (165 µg) syntaxin-1A. SUVs containing either t-SNARE or syntaxin-1A were formed by rapid β-OG dilution below the critical micelle concentration by adding 1.4 ml reconstitution buffer. For the quantification of lipid recovery, a 20-µl aliquot (GUV input) was removed and stored at −20 °C. The liposome suspension was desalted using a PD10 column (GE Healthcare) equilibrated with desalting buffer 1 (1 mM HEPES–KOH, pH 7.5, 1 wt/vol% glycerol, 10 µM EGTA–KOH, 1 mM DTT) and snap frozen in four aliquots in liquid nitrogen and stored in −80°C. These syntaxin-1A and t-SNARE SUVs were later used to prepare GUVs. The final protein to lipid ratios were determined by SDS–PAGE and Coomassie Blue staining of the proteins and Atto647 fluorescence intensity measurements of the lipids.

VAMP2/synaptotagmin-1-SUVs were prepared as described previously *Malsam et al., 2012*; *Weber et al., 1998* with the following modification: VAMP2 and synaptotagmin-1 were reconstituted at a protein to lipid ratio of 1:350 and 1:800, respectively. SUVs, which were used for the SNAP25 recruitment assay, were not dialyzed twice, but directly harvested after the density gradient flotation and lipid recovery and protein-to-lipid ratio were determined (Stx1-SUVs: 1:1900; Syt1/V2-SUVs: Syt1 1:700, VAMP2 1:300; V2-SUVs: VAMP2 1:200; Syt1-SUVs: Syt1 1:700).

## GUV preparation

GUVs were prepared as described previously (*Kedar et al., 2015*). Briefly, t-SNARE or syntaxin-1A liposomes (1.25 µmol lipid) were loaded onto a midi column (GE Healthcare) equilibrated with desalting buffer 2 containing trehalose (1 mM HEPES–KOH, pH 7.5, 0.5 wt/vol% glycerol, 10 µM EGTA–KOH, pH 7.4, 50 µM MgCl$_2$, 1 mM DTT, 10 mM trehalose). 1.4 ml eluate were collected and liposomes sedimented in a TLA-55 rotor (Beckman) at 35,000 rpm for 2 hr at 4°C. The pellet was resuspended by rigorous vortexing and, while vortexing, was diluted with 10 µl of pellet resuspension buffer (1 mM HEPES–KOH, pH 7.4, 10 µM EGTA–KOH, 50 µM MgCl$_2$) to lower the osmotic strength. The total volume (20–25 µl) was spread as a uniform layer (14 mm diameter) on the surface of a platinum foil (Alfa Aesar; 25 × 25 mm, 0.025 mm thick) attached to a glass slide as support. After drying the liposome suspension for 50 min at 50 mbar, the incubation chamber was assembled using an O-ring (2 × 18 mm), filled with 620 µl of swelling buffer (1 mM 4-(2-Hydroxyethyl)-1-piperazinepropanesulfonic acid (EPPS–KOH), pH 8.0, 240 mM sucrose [Ca$^{2+}$ free], 1 mM DTT) and closed using a second platinum plate. Conductive copper tape (3 M) was attached to the platinum foil to connect the assembly with a function generator (Voltcraft 8202). GUVs were generated by electro-formation at 10 Hz and 1 V at 0°C overnight.

## Lipid mixing assay

The membrane fusion assay was performed as described previously (*Malsam et al., 2012*), except that in the fusion buffer HEPES was replaced with 20 mM MOPS pH 7.4. Briefly, t-SNARE-GUVs (14 nmol lipid, 15 ± 0.7 pmol t-SNARE) were preincubated with or without 6 µM (0.6 nmol) CpxII for 5 min on ice in fusion buffer containing 0.1 mM EGTA–KOH and 0.5 mM MgCl$_2$. When using syntaxin-1A-GUVs (14 nmol lipid, 14 pmol syntaxin-1A), these preincubations contained 2 µM (0.2 nmol) of soluble SNAP25b in addition. Subsequently, VAMP2/Syt1-SUVs (2.5 nmol lipid, 4.5 pmol VAMP2, 2 pmol Syt1) were added to the GUV reaction mix resulting in 104 µl sample volume. After 10 min on ice, 100 µl of the GUV–SUV mixes were transferred into a prewarmed 96-well plate (37°C) and fluorescence emitted by Atto488 ($\lambda_{ex}$ = 485 nm, $\lambda_{em}$ = 538 nm) was measured in a Synergy 4 plate reader (BioTek Instruments GmbH) at intervals of 10 s. Ca$^{2+}$ was added to a final free concentration of 100 µM (https://somapp.ucdmc.ucdavis.edu/pharmacology/bers/maxchelator/CaEGTA-TS.htm) after 2 or 30 min for t-SNARE-GUVs or syntaxin-1A-GUVs, respectively. The fusion reactions were stopped after 4 min for t-SNARE-GUVs or after 1 hr for syntaxin-1A-GUVs by the addition of 0.7% SDS and 0.7% n-Dodecyl-β-D-Maltosid (DDM). The resulting 'maximum' fluorescent signal was used to normalize the fusion-dependent fluorescence. As a negative control, SUVs were treated with Botulinum NeuroToxin type D (BoNT-D) and their fluorescence signals were subtracted from individual measurement sets. Three independent fusion experiments were performed for each mutant.

## SUV–GUV-binding assay

All SUV–GUV-binding studies were performed in an ice bath and all pipetting steps were carried out in the cold room to avoid membrane fusion (*Parisotto et al., 2012*; *Malsam et al., 2012*; *Weber et al., 1998*). Before starting the incubation, potential SUV aggregates were removed by centrifugation. T-SNARE-GUVs (28 nmol lipid, 30 ± 1.4 pmol t-SNARE) were preincubated with VAMP2/Syt1-SUVs (5 nmol lipid, 9 pmol VAMP2, 4 pmol Syt1) on ice in 100 µl fusion buffer (20 mM MOPS–KOH, pH 7.4, 135 mM KCl, 1 mM DTT) with 0.1 mM EGTA and 1 mM MgCl$_2$. After 10 min incubation, the reactions were underlaid with 20 µl of a sucrose cushion (1 mM MOPS–KOH, pH 7.4, 60 mM sucrose, 1 mM DTT) and the GUVs with attached SUVs were re-isolated by centrifugation for 10 min. After removing the supernatant, the pellets (in 10 µl remaining volume) were resuspended, transferred into new tubes, treated with 100 µl of 1% SDS/1% DDM and the SUV recovery was determined by measuring the Atto488 fluorescence.

To determine the respective inputs, 28 nmol GUV lipids or 5 nmol SUV lipids were treated with 1% SDS/1% DDM (final) and the corresponding Atto647 and Atto488 fluorescence was measured at $\lambda_{ex}$ = 620/40 nm, $\lambda_{em}$ = 680/30 nm and $\lambda_{ex}$ = 485/20 nm, $\lambda_{em}$ = 528/20 nm, respectively. A sample lacking SUVs was used to determine the GUV recovery (usually 80–95%). GUV recovery of each sample was used to normalize the respective SUV docking. A sample without GUVs was used to determine the absolute background (usually <15%), which was subtracted from all samples.

## SNAP25 recruitment assay/SUV flotation assay

SUVs containing 35 pmol syntaxin-1A or 210 pmol VAMP2 and/or 100 pmol Synaptotagmin-1 (66 nmol lipid for Stx1-, V2/Syt1-, or Syt1-SUVs, 40 nmol lipid for V2-SUVs) were mixed with 180 pmol SNAP25 in a final assay volume of 50 µl in fusion buffer (20 mM MOPS–KOH, pH 7.4, 135 mM KCl, 1 mM DTT). After 2 hr on ice, allowing the SNARE complex formation, the samples were diluted five times with fusion buffer and mixed with the equivalent volume of 80% nycodenz solution. After overlaying the sample with 100 µl 35% nycodenz, 25 µl 20% nycodenz solution, and finally with 5 µl fusion buffer in an ultra-clear tube (5 × 41 mm), the liposomes were isolated by centrifugation for 3 hr 40 min at 55,000 rpm at 4°C in a SW 60 rotor (SW 60 Ti, Beckman). 20 µl were harvested from the top of the gradient and mixed with 8 µl of 4× Laemmli buffer (final 62.5 mM Tris–HCl, pH 6.8, 10% glycerol, 2% SDS, 50 mM β-mercaptoethanol, 0.1% bromphenol blue). 18 µl of this mixture were used to quantify the amount of recruited SNAP25b by SDS–PAGE followed by Coomassie Staining and Silver Staining. Using the Fiji software (ImageJ based) the Coomassie stained band intensity of syntaxin-1A or VAMP2 or Syt1, respectively, were determined and normalized to the respective mean. Subsequently, Silver Staining was used to quantify the band intensities of SNAP25b, and these values were normalized to the intensities of the relative protein (e.g. Syntaxin-1A) based on the Coomassie Staining. From this, the ratios between the SNAP25b mutants and WT were determined.

## Temperature-dependent dissociation of v-/t-SNARE complexes in SDS

SNARE complex stability was determined as described previously (*Schupp et al., 2016*). Briefly, preassembled full-length t-SNARE complexes (WT and mutants, 10 µM) were incubated with the cytoplasmic domain of VAMP2 (30 µM) in 25 mM MOPS (3-(Nmorpholino) propanesulfonic acid)–KOH, pH 7.4, 135 mM KCl, 1% Octyl β-D-glucopyranoside, 1 mM Dithiothreitol, 10 mM TECEP (Tris(2-carboxyethyl)phosphine hydrochloride)–KOH, pH 7.4, 1 mM EDTA–NaOH, pH 7.4 overnight at 0°C and subsequently for 1 hr at 25°C. Subsequently, 37 µl of reaction mixture (36 µg of total protein) was diluted with 213 µl of 1× Laemmli buffer. 7.5 µl aliquots were incubated at the indicated temperatures for 5 min. Samples were analyzed by SDS–PAGE (15% gels) and proteins were visualized by Coomassie brilliant blue staining. Temperature-dependent dissociation of the SNARE complex was quantified by the appearance of free syntaxin-1A (35 kDa protein band released from the high MW SNARE complex) using the Fiji (ImageJ based) software. Data were normalized to the maximum value of a measurement set.

## Silver Staining

Coomassie prestained gels from SDS–PAGE were destained overnight in destain solution (30% methanol, 10% acetic acid). Gels were gently washed 30 min with 10% ethanol and 1 min with 0.02% sodium thiosulfate. After short washing with deionized water, gels were stained (0.03% PFA, 0.002% silver nitrate) for 15 min and again quickly washed with water. Developing solution (0.06% sodium carbonate, 0.018% PFA, 0.0002% sodium thiosulfate) was applied to the gels until protein band intensities were satisfactorily stained. The reactions were stopped by replacing the staining solution with 0.07% acetic acid. Gels were scanned and quantified by using the Fiji (ImageJ based) software.

## Atomistic MD simulations

Simulations were carried out at the atomic level using classical MD. We used the CHARMM36m force field for protein, CHARMM TIP3P for water, and the standard CHARMM36 for ions (*Huang et al., 2017*). GROMACS 2022 software package was used for performing these simulations (*Abraham et al., 2015*). The 3D structure of SNAP25 WT protein was generated through AlphaFold2, utilizing the first ranked structure obtained using the default settings of ColabFold v1.5.2 (*Mirdita et al., 2022*). Both WT and the V48F and D166Y mutant variants' topology files and initial 3D structures, inclusive of water and ions, were produced via the CHARMM-GUI web interface (*Lee et al., 2016*). All systems were hydrated, neutralized with counter ions, and supplemented with 150 mM potassium chloride to replicate experimental conditions. Following the CHARMM-GUI recommended protocol, systems were energy-minimized, equilibrated under NpT conditions, temperature-stabilized at 310 K by the Nose–Hoover thermostat with a 1.0 ps time constant (*Evans and Holian, 1985*), and maintained a constant 1 atm pressure via the Parrinello–Rahman barostat, setting the time constant at 5.0 ps and an isothermal compressibility at $4.5 \times 10^{-5}$ bar$^{-1}$ (*Parrinello and Rahman, 1981*). Isotropic

pressure coupling was utilized in the simulations. The Verlet scheme determined neighbor searches, updating once every 20 steps (*Verlet, 1967*). Electrostatics were computed using the Particle Mesh Ewald method (*Darden et al., 1993*) with parameters set to a 0.12-nm spacing, a tolerance $10^{-5}$, and a 1.2-nm cutoff. Periodic boundary conditions were applied in all three dimensions. Simulations ran with a 2-fs timestep until 800 ns was achieved. As shown in *Figure 9—figure supplement 2* panel A, the structures used for protein alignment represent the predominant structure from the most populated cluster. Clustering was based on the RMSD value using the GROMACS 'gmx cluster' tool and the Gromos algorithm (*Daura et al., 1999*), setting an RMSD cutoff for neighbor structures at 0.6 nm (*Legrand et al., 2020*). The RMSD analysis was conducted using the average structure of the WT as a reference. Interaction energies, including both short-range Coulomb and Lennard–Jones forces, were computed throughout the 800 ns trajectory. The autocorrelation function of the data indicated that correlations diminished substantially after approximately 6 ns. Given this reduced correlation, we adopted a block size of 200 ns to ensure statistical independence between blocks.

All structures generated using AlphaFold2, as well as the initial structures and topology files for atomistic MD simulations, were deposited to ZENODO (https://doi.org/10.5281/zenodo.10051665).

## Statistical analysis

Graphs (bar and line) display mean ± standard error of the mean with all points displayed, except when otherwise noted; for electrophysiological experiments $n$ denotes the number of cells recorded and is given in the legends. For in vitro experiments the number of biological replicates was 3 unless stated otherwise in the legends. Statistics were performed using GraphPad Prism 9. Unless otherwise noted, statistical differences between several groups were determined by one-way analysis of variance (ANOVA); post-test was Dunnett's test comparing to the WT condition as a reference, unless otherwise noted. Equal variance of groups was tested by the Brown–Forsythe test; in case of a significant test, the Brown–Forsythe ANOVA test, which does not assume equal variances, was used instead. Kruskal–Wallis test was used in cases, where the data structure contained many identical values (zeros). Pairwise testing was carried out using unpaired $t$-test, or Welch's $t$-test in case of significantly different variances as determined by an $F$-test. The test is mentioned in the figure legend; if no test is mentioned, the difference was not significantly different. Significance was assumed when $p < 0.05$ and the level of significance is indicated by asterisks: *$p < 0.05$, **$p < 0.01$, ***$p < 0.001$, ****$p < 0.0001$.

## Materials availability

New materials generated during this project are available upon request.

## Acknowledgements

We would like to thank Dorte Lauritsen and Ursula Göbel for excellent technical assistance. This investigation was supported by the Novo Nordic Foundation (NNF19OC0058298, JBS), the Independent Research Fund Denmark (8020-00228 A, JBS), the Lundbeck Foundation (R277-2018-802, JBS), and the German Research Foundation DFG (grant 278001972 - TRR 186, THS, JM; LO 2821/1-1, FL; DFG SFB/TRR 186 A1, WN; DFG Ni 423/13-1, WN). The authors gratefully acknowledge the data storage service SDS@hd supported by the Ministry of Science, Research, and the Arts Baden-Württemberg (MWK), the German Research Foundation (DFG) through grants INST 35/1314-1 FUGG and INST 35/1503-1 FUGG. We acknowledge the computing resources provided by the CSC - IT Center for Science Ltd (Espoo, Finland).

## Additional information

### Funding

| Funder | Grant reference number | Author |
| --- | --- | --- |
| Novo Nordisk Foundation | NNF19OC0058298 | Jakob Balslev Sørensen |
| Independent Research Fund Denmark | 8020-00228A | Jakob Balslev Sørensen |

| Funder | Grant reference number | Author |
|---|---|---|
| Lundbeck Foundation | R277-2018-802 | Jakob Balslev Sørensen |
| Deutsche Forschungsgemeinschaft | 278001972 - TRR 186 | Jacqueline Murach Thomas H Söllner |
| Deutsche Forschungsgemeinschaft | LO 2821/1-1 | Fabio Lolicato |
| Deutsche Forschungsgemeinschaft | SFB/TRR 186 A1 | Walter Nickel |
| Deutsche Forschungsgemeinschaft | Ni 423/13-1 | Walter Nickel |
| Deutsche Forschungsgemeinschaft | INST 35/1314-1 FUGG | Fabio Lolicato |
| Deutsche Forschungsgemeinschaft | INST 35/1503-1 FUGG | Fabio Lolicato |
| CSC - IT Center for Science | | Fabio Lolicato |

The funders had no role in study design, data collection, and interpretation, or the decision to submit the work for publication.

## Author contributions

Anna Kádková, Maiken Østergaard, Data curation, Formal analysis, Investigation, Writing – original draft, Writing – review and editing; Jacqueline Murach, Data curation, Formal analysis, Investigation, Writing – review and editing; Andrea Malsam, Jörg Malsam, Formal analysis, Investigation, Writing – review and editing; Fabio Lolicato, Formal analysis, Investigation, Visualization, Writing – review and editing; Walter Nickel, Supervision, Funding acquisition, Writing – review and editing; Thomas H Söllner, Conceptualization, Supervision, Funding acquisition, Writing – review and editing; Jakob Balslev Sørensen, Conceptualization, Supervision, Funding acquisition, Writing – original draft, Project administration, Writing – review and editing

## Author ORCIDs

Anna Kádková http://orcid.org/0000-0001-6648-9679
Maiken Østergaard http://orcid.org/0009-0005-3869-9069
Fabio Lolicato http://orcid.org/0000-0001-7537-0549
Walter Nickel http://orcid.org/0000-0002-6496-8286
Jakob Balslev Sørensen https://orcid.org/0000-0001-5465-3769

## Ethics

Permission to keep, breed, and use Snap25 and Syt1 mice was obtained from the Danish Animal Experiments Inspectorate (2018-15-0202-00157) and followed institutional guidelines as overseen by the Institutional Animal Care and Use Committee (IACUC).

Reviewer #1 (Public Review): https://doi.org/10.7554/eLife.88619.3.sa1
Reviewer #2 (Public Review): https://doi.org/10.7554/eLife.88619.3.sa2
Author Response https://doi.org/10.7554/eLife.88619.3.sa3

# Additional files

## Supplementary files
• MDAR checklist

## Data availability

All data generated and used during this study are included in the manuscript and supporting files. Source data are included for all figures. All structures generated using AlphaFold2, as well as the initial structures and topology files for atomistic molecular dynamics simulations, were deposited to ZENODO (https://doi.org/10.5281/zenodo.10051665).

The following dataset was generated:

| Author(s) | Year | Dataset title | Dataset URL | Database and Identifier |
|---|---|---|---|---|
| Lolicato F | 2023 | Atomistic molecular dynamics simulations of SNAP25 in water | https://zenodo.org/records/10051665 | Zenodo, 10.5281/zenodo.10051665 |

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

# Appendix 1

## Appendix 1—key resources table

| Reagent type (species) or resource | Designation | Source or reference | Identifiers | Additional information |
|---|---|---|---|---|
| Strain, strain background (*M. musculus*) | CD1 | Experimental medicine, Panum Stable, University of Copenhagen | | |
| Genetic reagent (*M. musculus*) | *Synaptotagmin-1 (syt1)* null allele | **Geppert et al., 1994** | | PMID:7954835 |
| Genetic reagent (*M. musculus*) | *Snap25* null allele | **Washbourne et al., 2002** | | PMID:11753414 |
| Transfected construct (*M. musculus*) | p156rrl-EGFP-SNAP25b | **Delgado-Martínez et al., 2007** | Local identifier, #192 | PMID:17728451 See constructs for rescue experiments |
| Transfected construct (*M. musculus*) | p156rrl-EGFP-SNAP25b-I67N | Sørensen lab, this paper | Local identifier, #193 | See constructs for rescue experiments |
| Transfected construct (*M. musculus*) | p156rrl-EGFP-SNAP25b-V48F | Sørensen lab, this paper | Local identifier, #195 | See constructs for rescue experiments |
| Transfected construct (*M. musculus*) | p156rrl-EGFP-SNAP25b-D166Y | Sørensen lab, this paper | Local identifier, #212 | See constructs for rescue experiments |
| Transfected construct (*M. musculus*) | p156rrl-EGFP-SNAP25b-I67N/E183K/S187K/T190K/E194K | Sørensen lab, this paper | Local identifier, #209 | See constructs for rescue experiments |
| Gene (human) | Complexin II, CPLX2 | Uniprot | Q6PUV4, CPLX2_HUMAN | |
| Gene (mouse) | VAMP2 | Uniprot | P63044, VAMP2_MOUSE | |
| Gene (rat) | Synaptotagmin-1 | Uniprot | P21707, SYT1_RAT | |
| Gene (rat) | Syntaxin-1A | Uniprot | P32851, STX1A_RAT | |
| Gene (mouse) | SNAP25B | Uniprot | P60879, SNP25_MOUSE | |
| Gene (human) | Complexin II, CPLX2 | Uniprot | Q6PUV4, CPLX2_HUMAN | |
| Strain, strain background (*Escherichia coli*) | BL21(DE3) | Agilent | Cat# 200131 | |
| Strain, strain background (*Escherichia coli*) | BL21(DE3)codon+ | Agilent | Cat# 230240 | |
| Recombinant DNA reagent | Complexin II | **Malsam et al., 2012** | Local identifier, pMDL80 | PMID:22705946 |
| Recombinant DNA reagent | VAMP2 | **Kedar et al., 2015**. | Local identifier, pSK28 | PMID:26490858 |
| Recombinant DNA reagent | VAMP2cd | **Ruiter et al., 2019** | Local identifier, pSK74 | PMID:30811985 |
| Recombinant DNA reagent | Synaptotagmin-1 | **Mahal et al., 2002** | Local identifier, pLM6 | PMID:12119360 |
| Recombinant DNA reagent | Syntaxin-1A | Söllner lab, this paper | Local identifier, pSK270 | See constructs for in vitro protein expression |
| Recombinant DNA reagent | SNAP25B | **Parlati et al., 1999** | Local identifier, pFP247 | PMID:11001058 |
| Recombinant DNA reagent | tSNARE | **Parlati et al., 1999** | Local identifier, pTW34 | PMID:11001058 |
| Recombinant DNA reagent | SNAP25B I67N | Söllner lab, this paper | Local identifier, pUG1 | See constructs for in vitro protein expression |

*Appendix 1 Continued on next page*

*Appendix 1 Continued*

| Reagent type (species) or resource | Designation | Source or reference | Identifiers | Additional information |
|---|---|---|---|---|
| Recombinant DNA reagent | SNAP25B V48F | Söllner lab, this paper | Local identifier, pUG2 | See constructs for in vitro protein expression |
| Recombinant DNA reagent | SNAP25B D166Y | Söllner lab, this paper | Local identifier, pUG3 | See constructs for in vitro protein expression |
| Recombinant DNA reagent | tSNARE SNAP25B I67N | Söllner lab, this paper | Local identifier, pUG7 | See constructs for in vitro protein expression |
| Recombinant DNA reagent | tSNARE SNAP25B V48F | Söllner lab, this paper | Local identifier, pUG8 | See constructs for in vitro protein expression |
| Recombinant DNA reagent | tSNARE SNAP25B D166Y | Söllner lab, this paper | Local identifier, pUG9 | See constructs for in vitro protein expression |
| Sequence-based reagent | SNAP25B I67N<br>fw: ttctttcatgtccttattgttttggtccatcccttcctc<br>rv: gaggaagggatggaccaaaacaataaggacatgaaagaa | Söllner lab, this paper | | Quick-change primer to generate SNAP25B I67N |
| Sequence-based reagent | SNAP25B V48F<br>fw: cttgctcatccaacataaacaaagtcctgatgccagc<br>rv: gctggcatcaggactttgtttatgttggatgagcaag | Söllner lab, this paper | | Quick-change primer to generate SNAP25B V48F |
| Sequence-based reagent | SNAP25B D166Y<br>fw: ctcattgcccatgtatagagccatatggcggagg<br>rv: cctccgccatatggctctatacatgggcaatgag | Söllner lab, this paper | | Quick-change primer to generate SNAP25B D166Y |
| Antibody | Anti-VGlut1 (guinea pig polyclonal) | Merck Millipore | Cat# AB5905<br>RRID: AB_2301751 | 1:1000; overnight at 4°C |
| Antibody | Anti-MAP2 (chicken polyclonal) | Abcam | Cat# Ab5392<br>RRID: AB_2138153 | 1:500; overnight at 4°C |
| Antibody | Anti-guinea pig Alexa Fluor 647 (goat polyclonal) | Thermo Fisher Scientific | Cat# A-21450<br>RRID: AB_2535867 | 1:4000; 1 hr at room temperature |
| Antibody | Anti-chicken Alexa Fluor 568 (goat polyclonal) | Thermo Fisher Scientific | Cat# A11041<br>RRID: AB_2534098 | 1:1000; 1 hr at room temperature |
| Antibody | Anti-SNAP25 (mouse monoclonal) | Synaptic Systems | Cat# 111011<br>RRID: AB_887794 | 1:10,000; overnight at 4°C |
| Antibody | Anti-VCP (mouse monoclonal) | Abcam | Cat# Ab11433<br>RRID: AB_298039 | 1:2000; overnight at 4°C |
| Antibody | Anti-VCP (rabbit monoclonal) | Abcam | Cat# Ab109240<br>RRID: AB_10862588 | 1:5000 overnight at 4°C |
| Antibody | Anti-mouse HRP (polyclonal goat) | Agilent (Dako) | Agilent, cat# P044701-2<br>RRID: AB_2617137 | 1:10,000; 1 hr at room temperature |
| Antibody | Anti-rabbit HRP (polyclonal goat) | Agilent (Dako) | Agilent, cat# P044801-2<br>RRID: AB_2617138 | 1:10,000; 1 hr at room temperature |
| Commercial assay or kit | QuikChange II XL kit | Agilent | Agilent, cat# 200521 | |
| Commercial assay or kit | QIAprep Spin Miniprep Kit | QIAGEN | QIAGEN, cat# 27106 | |
| Commercial assay or kit | BCA Protein assay kit | Pierce | Pierce, cat# 23227 | |
| Commercial assay or kit | QuikChange site-directed DNA mutagenesis kit | Agilent | Cat# 200519 | |
| Commercial assay or kit | Venor GeM OneStep mycoplasma test | Minerva biolabs | Art. Nr. 11-8025 | |
| Chemical compound, drug | NaCl | Sigma-Aldrich | Sigma-Aldrich, cat# S9888 | |
| Chemical compound, drug | KCl | Sigma-Aldrich | Sigma-Aldrich, cat# P5405 | |
| Chemical compound, drug | Glucose | Sigma-Aldrich | Sigma-Aldrich, cat# G8270 | |

*Appendix 1 Continued on next page*

*Appendix 1 Continued*

| Reagent type (species) or resource | Designation | Source or reference | Identifiers | Additional information |
|---|---|---|---|---|
| Chemical compound, drug | $CaCl_2$ | Sigma-Aldrich | Sigma-Aldrich, cat# 31307 | |
| Chemical compound, drug | $MgCl_2$ | Sigma-Aldrich | Sigma-Aldrich, cat# M2393 | |
| Chemical compound, drug | 96% ethanol | VWR | VWR, cat# 20824.321 | |
| Chemical compound, drug | Sucrose | Sigma-Aldrich | Sigma-Aldrich, cat# S1888 | |
| Chemical compound, drug | EGTA | Sigma-Aldrich | Sigma-Aldrich, cat# E4378 | |
| Chemical compound, drug | HEPES | Sigma-Aldrich | Sigma-Aldrich, cat# H4034 | |
| Chemical compound, drug | Na-ATP | Sigma-Aldrich | Sigma-Aldrich, cat# A2383 | |
| Chemical compound, drug | Creatine phosphate | Sigma-Aldrich | Sigma-Aldrich, cat# P7936 | |
| Chemical compound, drug | Creatine phosphokinase | Sigma-Aldrich | Sigma-Aldrich, cat# C3755 | |
| Chemical compound, drug | Albumin | Sigma-Aldrich | Sigma-Aldrich, cat# A9418 | |
| Chemical compound, drug | Trypsin-inhibitor | Sigma-Aldrich | Sigma-Aldrich, cat# T9253 | |
| Chemical compound, drug | Penicillin/streptomycin | Gibco | Gibco, cat# 15140122 | |
| Chemical compound, drug | Fetal bovine serum | Gibco | Gibco, cat# 10500064 | |
| Chemical compound, drug | MEM non-essential amino acids (100×) | Gibco | Gibco, cat# 11140050 | |
| Chemical compound, drug | Collagen Type I | Corning | Corning, cat# 354236 | |
| Chemical compound, drug | Agarose Type II-A | Sigma-Aldrich | Sigma-Aldrich, cat# A-9918 | |
| Chemical compound, drug | Trypsin–EDTA (10×) | Gibco | Gibco, cat# 15090-046 | |
| Chemical compound, drug | Trypsin–EDTA (1×) | Gibco | Gibco, cat# 25300-054 | |
| Chemical compound, drug | DMEM (1×) + GlutaMAX-1 | Gibco | Gibco, cat# 31966-021 | |
| Chemical compound, drug | Geneticin Selective Antibiotic (G418 Sulfate) | Gibco | Gibco, cat. 11811064 | |
| Chemical compound, drug | Poly-D-lysine | Sigma-Aldrich | Sigma-Aldrich, cat# P7405 | |
| Chemical compound, drug | Glacial acetic acid | Sigma-Aldrich | Sigma-Aldrich, cat# 695084 | |
| Chemical compound, drug | Neurobasal | Gibco | Gibco, cat# 21103049 | |
| Chemical compound, drug | Neurobasal-A | Gibco | Gibco, cat# 10888022 | |
| Chemical compound, drug | HBSS | Gibco | Gibco, cat# 24020-091 | |

*Appendix 1 Continued*

| Reagent type (species) or resource | Designation | Source or reference | Identifiers | Additional information |
|---|---|---|---|---|
| Chemical compound, drug | B-27 supplement | Gibco | Gibco, cat# 17504044 | |
| Chemical compound, drug | 100× Glutamax-1 supplement | Gibco | Gibco, cat# 35050-061 | |
| Chemical compound, drug | β-Mercaptoethanol | Sigma-Aldrich | Sigma-Aldrich, cat# M7522 | |
| Chemical compound, drug | Paraformaldehyde | Sigma-Aldrich | Sigma-Aldrich, cat# P6148 | |
| Chemical compound, drug | PIPES | Sigma-Aldrich | Sigma-Aldrich, cat# 80635 | |
| Chemical compound, drug | Triton X-100 | Sigma-Aldrich | Sigma-Aldrich, cat# T8787 | |
| Chemical compound, drug | Octyl-beta-glucoside | Thermo Fisher Scientific | Thermo Scientific, cat# 28310 | |
| Chemical compound, drug | BSA | Sigma-Aldrich | Sigma-Aldrich, cat# A4503 | |
| Chemical compound, drug | Protease cocktail inhibitor | Thermo Scientific | Thermo Scientific, cat# 87785 | |
| Chemical compound, drug | RIPA buffer | Sigma-Aldrich | Sigma-Aldrich, cat# R0278 | |
| Chemical compound, drug | NuPAGE MES SDS Running Buffer | Invitrogen | Invitrogen, cat# NP0002 | |
| Chemical compound, drug | Trizma base | Sigma-Aldrich | Sigma-Aldrich, cat# T1503 | |
| Chemical compound, drug | Glycine | Sigma-Aldrich | Sigma-Aldrich, cat# G8898 | |
| Chemical compound, drug | 10% SDS | Sigma-Aldrich | Sigma-Aldrich, cat# 71736 | |
| Chemical compound, drug | Tween20 | Sigma-Aldrich | Sigma-Aldrich, cat# P9416 | |
| Chemical compound, drug | Sample Reducing Agent | Invitrogen | Invitrogen, cat# B0009 | |
| Chemical compound, drug | LDS Sample Buffer | Invitrogen | Invitrogen, cat# B0007 | |
| Chemical compound, drug | Difco Skim Milk | BD Life Sciences | BD Life Sciences, cat# 232100 | |
| Chemical compound, drug | ECL plus Western blotting substrate | Pierce | Pierce, cat# 32132 | |
| Chemical compound, drug | POPE (1-hexadecanoyl-2-octadecenoyl-*SN*-glycero-3-phosphoethanolamine) | Avanti Polar Lipids | Cat# 850757 P-25 mg | |
| Chemical compound, drug | POPC (1-palmitoyl-2-oleoyl-*SN*-glycero-3- phosphocholine) | Avanti Polar Lipids | Cat# 850457 P-25 mg | |
| Chemical compound, drug | DOPS (1,2-dioleoyl-*SN*-glycero-3-phosphoserine) | Avanti Polar Lipids | Cat# 840035 P-10 mg | |
| Chemical compound, drug | Cholesterol (from ovine wool) | Avanti Polar Lipids | Cat# 700000 P-100 mg | |
| Chemical compound, drug | PI (L-α-phosphatidylinositol) | Avanti Polar Lipids | Cat# 840042 P-25 mg | |
| Chemical compound, drug | PI(4,5)P2 (L-α-phosphatidylinositol-4,5-bisphosphate) | Avanti Polar Lipids | Cat# 840046 P-1 mg | |

*Appendix 1 Continued on next page*

*Appendix 1 Continued*

| Reagent type (species) or resource | Designation | Source or reference | Identifiers | Additional information |
|---|---|---|---|---|
| Chemical compound, drug | Atto647-DPPE (1,2-dipalmitoyl-*SN*-glycero-3-phosphoethanolamine) | ATTO-TEC | Cat# AD 647 N-151 | |
| Chemical compound, drug | Atto488-DPPE (1,2-dipalmitoyl-*SN*-glycero-3-phosphoethanolamine) | ATTO-TEC | Cat# AD 488-151 | |
| Chemical compound, drug | Atto550-DPPE (1,2-dipalmitoyl-*SN*-glycero-3-phosphoethanolamine) | ATTO-TEC | Cat# AD 550-151 | |
| Chemical compound, drug | Nycodenz | Axis-Shield | Prod. No. 18003 | |
| Other | MonoS 5/50 GL column; MonoQ 5/50 GL column | GE Healthcare | Discontinued | Ion exchange columns |
| Other | Protino Ni-NTA Agarose | Macherey-Nagel | Cat# 745400 | Affinity resin |
| Other | PreScission protease | Cytiva | Cat# 27084301 | Site-specific protease |
| Other | PD10 desalting column PD MidiTrap G-25 column | Cytiva | Cat# 17085101 Cat# 28918008 | Buffer exchange columns |
| Other | Platinum foil 25 × 25 mm | Fisher scientific | Cat# 11356429 | Component of the electrode assembly for GUV formation |
| Other | Copper tape, 25 mm | 3M | Cat# ET 1181 | Component of the electrode assembly for GUV formation |
| Other | PTFE tape, 25.4 mm | 3M | Cat# 5491 | Component of the electrode assembly for GUV formation |
| Software, algorithm | Igor | Wavemetrics | | |
| Software, algorithm | Patchmaster v2.73 | HEKA | | |
| Software, algorithm | MiniAnalysis v6.0.7 | Synaptosoft | | |
| Software, algorithm | Zeiss Zen Blue | | | |
| Software, algorithm | Zeiss Zen Black | | | |
| Software, algorithm | SynD Automated Image Analysis | *Schmitz et al., 2011* | PMID:21167201 | |
| Software, algorithm | GraphPad Prism 9 | | | |
| Software, algorithm | ImageJ | NIH software | | |
| Software, algorithm | | | | |
| Software, algorithm | GROMACS, version 2022 | GROMACS | | |
| Software, algorithm | AlphaFold, version 2 | *Jumper et al., 2021* | PMID:34265844 | |
| Software, algorithm | ColabFold, version 1.5.2 | *Mirdita et al., 2022* | PMID:35637307 | |

