## [Editor Report · eLife assessment]

This study documents **important** findings on three variants in SNAP25 that are associated with developmental and epileptic encephalopathy. The thorough characterization of synaptic release and in vitro vesicle fusion phenotypes provides interesting information about the nature of the SNAP25 variants. The evidence supporting the claims is **compelling**, and this work will be of interest to neuroscientists working on SNAP25, SNAP25-associated encephalopathy, and synaptic vesicle exocytosis.

---

## [Referee Report · Reviewer #1 (Public Review)]

The manuscript by Kadkova et al. describes an electrophysiological analysis of 3 neurodevelopmental disease-causing SNAP-25 mutations in hippocampal neuron autaptic cultures. The work expands on a prior study of these 3 mutations, along with several others in SNAP-25, that was performed in acutely dissociated hippocampal cultures by another group (Alten et al, 2021). Most of the physiology defects found are pretty similar for the 3 mutations the two research groups characterized, with differences largely found in the effects on the size of the readily releasable pool (RRP) of SVs. These differences could be due to technical differences in the approach but are also likely to reflect in part differences in autapses as a model that have been previously described. In addition to the physiological analysis in cultured neurons, the current work extends the analysis beyond the prior study by analyzing the effects of these SNAP-25 mutations in in vitro liposome fusion assays with purified proteins, and some modeling of the effects on energy landscapes during priming and fusion.

The authors use lentiviral expression of wildtype or one of the 3 mutants in SNAP-25 autaptic neurons and assay neuronal survival and synaptic output. The authors also combine wildtype with each of the 3 mutants as well, given these diseases manifest as spontaneous mutations in only 1 of the SNAP-25 alleles, suggesting a dominant effect. The authors observe that the V48F and D166Y alleles (that are suggested to disrupt the Syt1-SNAP-25/SNARE interface) result in a very large increase in spontaneous release that exceeds the Syt1 null mutant alone, suggesting an effect on spontaneous SV release beyond a lack of Syt1 regulation of SNARE-mediated fusion. In contrast, Syt1 nulls have a much more severe loss of evoked release, through both V48F and D166Y also have modest decreases in release. They find both mutants also cause a decrease in the RRP. Applying some modeling for these results, the authors suggest V48F and D166Y lowers the energy barrier for fusion, creating the enhanced spontaneous release rates and causing a decrease of the RRP. They also find evidence for reduced SV priming. In contrast, a SNAP-25 I167N disease mutation in the SNARE assembly interaction layer causes dramatic decreases in both evoked and spontaneous release, consistent with a disruption to SNARE assembly/stability. In vitro fusion assays with these mutant SNAP-25 alleles was also done and provided supportive evidence for these interpretations for all 3 alleles. The ability to control calcium, Syt1, PIP2 and Complexin levels in the in vitro assays provided additional information on defining the precise steps of the fusion process these mutations disrupt. Together, the study indicates the I167N mutation acts as a dominant-negative allele to block fusion, while the other two alleles have both loss- and gain-of function properties that cause more complex disruptions that decrease evoked release while dramatically enhancing spontaneous fusion.

Overall, these results build on prior work and shed light on how disruptions to the SNAP-25 t-SNARE alter the process of SV priming and fusion.

---

## [Referee Report · Reviewer #2 (Public Review)]

Kádková, Murach, Pedersen, and colleagues studied how three disease-causing missense mutations in SNAP25 affect synaptic vesicle exocytosis. These mutations have previously been studied by Alten et al., 2021. The authors observed similar impairments in spontaneous and evoked release as Alten et al., 2021, but the measurement of readily releasable pool (RRP) size differed between the two studies. The authors found that the V48F and D166Y mutations affect the interaction with the Ca2+ sensor synaptotagmin-1 (Syt1), but do not entirely phenocopy Syt1 loss-of-function because they also exhibit a gain-of-function. Thus, these mutations affect multiple aspects of the energy landscape for vesicle priming and fusion. The I67N mutation specifically increases the fusion energy barrier without affecting upstream vesicle priming.

The strength of the study includes careful and technically excellent dissection of the synaptic release process and a combination of electrophysiology, biophysics, and modeling approaches. This study gained a deeper mechanistic understanding of these mutations in vesicle exocytosis than the previous study but did not result in a paradigm shift in our understanding of SNAP25-associated encephalopathy because the same spontaneous and evoked release phenotypes were previously identified.

Comments on revised version:

The authors fully addressed the two previous technical concerns and improved the introduction of the paper.

---

## [Author Response]

The following is the authors’ response to the original reviews.

We would like to thank the reviewers for their work, and the very useful comments.

**Public reviews:**

**Reviewer #2**
1. The authors discussed possible reasons for the different results of the RRP sizes between this study and Alten et al., 2021. One of them is how the hypertonic solution is applied. The authors thought that the long application of hypertonic solution in Alten et al., 2021 caused an overlapping release of RRP and upstream vesicle pools because Alten et al., 2021 measured 10-fold larger RRP size than what was measured in this study. However, Alten et al., 2021 measured RRP from IPSCs and a single inhibitory vesicle fusion causes larger charge transfer than an excitatory vesicle. The authors need to take this into consideration and 10-fold is likely an overestimate.

Answer: Thank you for pointing out this important difference. We have modified the text in the Discussion accordingly and we no longer refer to the 10-fold difference.

1. Statistical tests should be performed for protein expression levels (Fig 2A and Fig 10A) and in vitro fusion assays (Fig 8D,E and Fig 9 B,C).

Answer: We inserted new panels B and C in Fig. 2 and Fig. 10 showing all the Western Blot data and performed statistical tests (none were significant). For the in vitro fusion assays, we have inserted statistical tests in panels 8E and 9C. The quantities in those panels (subdivided into “Pre Ca2+”, “post Ca2+” and “end fusion”) are based on the data in Figure 8D and 9B. We have therefore not inserted separate statistical tests in Figures 8D and 9B.

**Reviewer #1 (Recommendations For The Authors):**
It would be quite interesting for future studies to address how these three mutations in SNAP-25 behave in the Syt1 null background in their electrophysiological experiments. Does the I167N allele block the enhanced spontaneous release in the Syt1 null? Do the V48F and D1667 alleles synergize with Syt1 to enhance spontaneous release to even higher levels? By examining how different components interact to shape the energy landscape for priming and fusion, these types of approaches should be quite revealing.

Answer: We agree with the reviewer that these future studies would be interesting. Unfortunately, they are beyond our current capacities.

**Reviewer #2 (Recommendations For The Authors):**
1. In the introduction, when discussing haploinsufficiency of Munc18-1 causes a decrease in release, additional references should be included, for example, the studies in flies (Wu et al., 1998, EMBO), human neurons (Patzke et al., 2015 JCI), and mouse neurons (Toonen et al., 2006 PNAS; Chen et al., 2020 eLife).

Answer: Thank you for the suggestion. We have rewritten the text and added additional references.

1. The authors may consider introducing additional motivations and significance of this study. For example, the evoked EPSCs cannot be properly measured in the cultures of Alten et al., 2021, but was properly studied here.

Answer: We agree and have added additional motivations in the Introduction.